# Defending Deep Neural Networks against Backdoor Attacks via Module Switching

## Abstract

The exponential increase in Deep Neural Networks (DNNs) parameters has significantly raised the cost of independent training, particularly for resource-constrained entities, leading to a growing reliance on open-source models. However, the opacity of these training processes exacerbates security risks, making these models more susceptible to malicious threats, such as backdoor attacks, while also complicating defense strategies. Merging homogeneous models has emerged as a cost-effective post-training defense. Current approaches, such as weight averaging, only partially mitigate the impact of poisoned parameters and are largely ineffective in disrupting the pervasive spurious correlations embedded across model parameters. To address this, we propose a novel module-switching strategy and validate its effectiveness both theoretically and empirically on two-layer networks, showing its remarkable ability to break spurious correlations and achieve higher backdoor divergence than weight averaging. For deep learning models, we further design and develop evolutionary algorithms to optimize fusion strategies, along with selective mechanisms to identify the most effective combination. Experimental results demonstrate that our defense exhibits strong resilience against backdoor attacks in both text and vision tasks, even when merging only a couple of compromised models.

## 1 Introduction

Deep neural networks (DNNs) draw much of their ability to learn from heterogeneous, real-world data. Although this diversity contributes to their remarkable performance across various tasks [4, 8, 48], it also leaves adversaries opportunities to implant carefully crafted patterns into training data, enabling malicious attacks. In particular, backdoor attacks poison a (small) portion of training samples with deceptive but stealthy triggers [6, 14]. As a result, the trained model behaves normally on 'clean' inputs but produces attacker-specified predictions when triggers appear at test time. The stealthiness of backdoor attacks raises serious security concerns and motivates effective defense research.

Recent advances in backdoor defense span both *training-phase* and *test-phase* approaches. However, many existing methods face significant practical constraints: (1) growing reliance on unverified models from open platforms (*e.g.,* HuggingFace) makes the training process and assets opaque; (2) increasingly stealthy backdoor triggers (*e.g.,* invisible syntactic patterns [39]) hinder effective data filtering and trigger inversion; (3) auxiliary datasets required for purification are not always available [68]; and (4) re-tuning incurs additional computational overhead [67].

Model combination techniques, such as model merging [19, 33], originally proposed for knowledge aggregation, have emerged as cost-effective defenses against backdoor attacks. For example, merging multiple compromised models can suppress textual backdoors [2]. However, naive weight averaging can still retain malicious behavior: merging a benign model with a compromised one may transfer the backdoor, while merging two poisoned models may preserve both backdoors [60]. An alternative

strategy seeks to combine models selectively, guided by trusted criteria, curated datasets, or reliable proxy models. For instance, Yang et al. [60] utilize perturbation methods associated with backdoor behaviors to iteratively mask related parameters, while Chen et al. [5] use auxiliary reference models to resolve information conflicts. Unfortunately, such trusted resources are not always available, and the reliability of newly identified resources is also questionable. Recent work [29, 45] shows that even compromised models can be leveraged to directly mitigate target backdoors, although there remain risks that the auxiliary model could introduce additional backdoor threats.

We propose *Module Switching*, a defense framework that selectively exchanges network modules among models trained on related domains. The intuition rests on the observation that backdoor attacks introduce "shortcuts" within DNNs, exploiting spurious correlations to trigger malicious behavior [13, 17, 61]. Because different attacks create distinct shortcuts, disrupting these pathways by swapping modules may effectively mitigate the corresponding vulnerabilities, as shown in Figure 1.

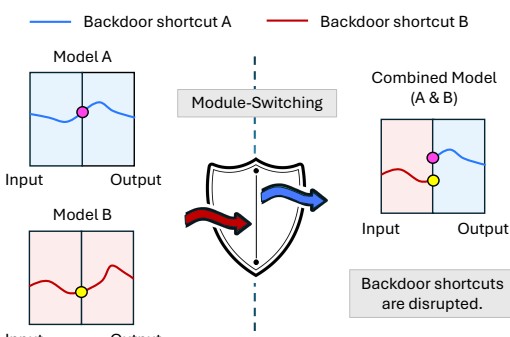

Figure 1: An illustration of Module-Switching Defense (MSD). By switching weight modules between compromised models (*left*), the spurious correlations (shortcuts) learned from backdoored tasks are effectively disrupted in the combined model (*right*).

Identifying every shortcut is computationally challenging due to numerous parameter interactions and the requirement of extra data. We therefore reformulate shortcut disruption as an optimization problem, searching for an effective module-switching strategy that breaks shortcut connections within a given model architecture. By combining heuristic scoring and an evolutionary algorithm, we obtain an index table that specifies which source model should fill each module slot. Since this module-switching scheme relies solely on architectural information, it generalizes across tasks and is transferable to any models sharing the same structure (*e.g.,* one strategy applicable to both *RoBERTa* [32] and *DeBERTa* [16]).

Our **Module-Switching Defense (MSD)** applies the strategy by assigning each module across the network a source-model index and recombining the selected modules to construct candidate models. Then, we identify the most robust candidate by comparing their representations on a small clean validation set (requiring only as few as 20–50 samples per class and no poisoned data). Because MSD is structure-driven, it is task-agnostic, counters a wide spectrum of backdoor threats, and preserves utility for downstream tasks. Our key contributions are as follows.

- We conduct an interpretable study on shallow networks, showing that module switching manages to effectively mitigate backdoor patterns while preserving semantics (Section 3).

- We propose and develop an MSD pipeline, which (1) establishes heuristic rules (Section 4.2) to guide an evolutionary algorithm search strategies that disrupt backdoor-related spurious correlations (Section 4.3), and (2) defines a feature-distance criterion to select the best model combination candidate (Section 4.4).

- We validate our method on DNNs in text and vision domains, showing it effectively mitigates various backdoor attacks, even when combining only compromised models (Section 5).

## 2   Related Work

**Backdoor Attacks.**   Backdoor attacks implant hidden vulnerabilities in DNNs, activating only when specific triggers appear in the input while maintaining normal behavior on benign data. They can be broadly categorized into two types in accordance with implanting methods: (1) *Data-poisoning attacks* inject trigger patterns into a small portion of the datasets with manipulated labels to train compromised models. Since being first discovered by Gu et al. [14], these attacks have evolved with diverse trigger designs in both vision [27, 36, 58] and text domains [7, 22, 39, 40]. In contrast, (2) *Weight-poisoning attacks* directly modify model weights to embed backdoors [10, 22]. The backdoor attacks can be considered correlating trigger patterns with predefined predictions in machine learning models, activated in inference [13, 17]. Our work focuses on defending against *data-poisoning attacks* in both text and vision domains, given their widespread adoption and potential risks.

**Backdoor Defense.** Backdoor defenses are typically classified by their deployment stage into (1) *training-phase* and (2) *test-phase* methods. *Training-phase* defenses treat poisoned data as outliers, aiming at detecting and removing them based on distinctive activation or learning patterns [17, 18, 25]. *Test-phase* defenses operate on inputs or model itself: data-level approaches reverse-engineer triggers [49] or filter anomalies [37], while model-level strategies detect trojaned models [31, 44, 50, 52] or purify models through pruning [30, 57, 67, 68] or unlearning [26, 56, 65]. While traditional model purification demands proxy data and additional retraining, recent research has focused on model combination strategies that require fewer assumptions and lower computational costs [2, 5, 29, 45, 60]. Building on this line of research, we propose a model confusion approach that reduces dependency on trusted resources while mitigating threats by disrupting spurious correlations in constituent models.

## 3 Module Switching in Two-layer Neural Networks

We theoretically and empirically examine whether *module switching* in two-layer networks can disrupt backdoor patterns introduced during fine-tuning, while preserving pretrained semantics. We find that swapping layer weights leads to greater deviation from backdoor patterns than weight averaging (WAG) [2, 51], yielding improved robustness against backdoored inputs.

**Setup and Notation.** We consider two-layer networks defined as $f(\boldsymbol{x}; \theta) = \boldsymbol{W}_2 \, \sigma(\boldsymbol{W}_1 \boldsymbol{x})$, with input $\boldsymbol{x} \in \mathbb{R}^N$ and parameters $\theta := \{\boldsymbol{W}_1, \boldsymbol{W}_2\}$, and activation function $\sigma(\cdot)$ (linear or non-linear). Training progresses in two stages: a *pretraining* stage, where shared weights $\boldsymbol{W}_1 \in \mathbb{R}^{K \times N}$ and $\boldsymbol{W}_2 \in \mathbb{R}^{N \times K}$ learn general semantics, followed by a *fine-tuning* stage that introduces updates ($\Delta \boldsymbol{W}_1^*$ and $\Delta \boldsymbol{W}_2^*$) to encode backdoor behavior in individual models $\mathcal{M}^*$.

In a linear network with identical activation, the fine-tuned model is $\mathcal{M}(\boldsymbol{x}) = (\boldsymbol{W}_2 + \Delta \boldsymbol{W}_2^*)(\boldsymbol{W}_1 + \Delta \boldsymbol{W}_1^*)\boldsymbol{x}$, which expands to a semantic term $\boldsymbol{S} = \boldsymbol{W}_2 \boldsymbol{W}_1$ and a backdoor component

$$\boldsymbol{B}^* = \boldsymbol{W}_2 \Delta \boldsymbol{W}_1^* + \Delta \boldsymbol{W}_2^* \boldsymbol{W}_1 + \epsilon^*, \tag{1}$$

such that $\mathcal{M}^*(\boldsymbol{x}) = (\boldsymbol{S} + \boldsymbol{B}^*)\boldsymbol{x}$, where the $\epsilon$-term $\epsilon^* = \Delta \boldsymbol{W}_2^* \Delta \boldsymbol{W}_1^*$ is a second-order interaction. It is typically much smaller in magnitude than first-order terms (*i.e.,* $\boldsymbol{W}_2 \Delta \boldsymbol{W}_1^* + \Delta \boldsymbol{W}_2^* \boldsymbol{W}_1$). We empirically verify this in Appendix C, and accordingly omit the $\epsilon$-term in subsequent analysis.

**Definition 1** (Weight-Averaged Model). *Let $i$ and $j$ index two fine-tuned backdoor models. Averaging the weights of $\mathcal{M}^i$ and $\mathcal{M}^j$ defines the* Weight-Averaged (WAG) *model [2], with parameters:*

$$\theta^{\text{wag}} := \left\{ \frac{1}{2} \left( \boldsymbol{W}_1 + \Delta \boldsymbol{W}_1^i \right) + \frac{1}{2} \left( \boldsymbol{W}_1 + \Delta \boldsymbol{W}_1^j \right), \frac{1}{2} \left( \boldsymbol{W}_2 + \Delta \boldsymbol{W}_2^i \right) + \frac{1}{2} \left( \boldsymbol{W}_2 + \Delta \boldsymbol{W}_2^j \right) \right\}.$$

*Assuming a linear network as above, we decompose the model as $\mathcal{M}^{\text{wag}}(\boldsymbol{x}) = (\boldsymbol{S} + \boldsymbol{B}^{\text{wag}}) \boldsymbol{x}$, where $\boldsymbol{S}$ denotes the shared pretrained semantic component, and the backdoor component is equivalent to*

$$\boldsymbol{B}^{\text{wag}} = \frac{1}{2} \boldsymbol{W}_2 \left( \Delta \boldsymbol{W}_1^i + \Delta \boldsymbol{W}_1^j \right) + \frac{1}{2} \left( \Delta \boldsymbol{W}_2^i + \Delta \boldsymbol{W}_2^j \right) \boldsymbol{W}_1.$$

**Definition 2** (Distance between Outputs from WAG and Constituent Models). *Under identity activation, $\ell_2$ distances between the WAG model and the two constituent models $\mathcal{M}^i$ and $\mathcal{M}^j$ are:*

$$\|\mathcal{D}^{\text{wag},i}\| = \|\mathcal{M}^{\text{wag}}(\boldsymbol{x}) - \mathcal{M}^i(\boldsymbol{x})\| = \frac{1}{2}\|\left( \boldsymbol{W}_2(\Delta \boldsymbol{W}_1^j - \Delta \boldsymbol{W}_1^i) + (\Delta \boldsymbol{W}_2^j - \Delta \boldsymbol{W}_2^i)\boldsymbol{W}_1 \right) \boldsymbol{x}\|,$$

$$\|\mathcal{D}^{\text{wag},j}\| = \|\mathcal{M}^{\text{wag}}(\boldsymbol{x}) - \mathcal{M}^j(\boldsymbol{x})\| = \frac{1}{2}\|\left( \boldsymbol{W}_2(\Delta \boldsymbol{W}_1^i - \Delta \boldsymbol{W}_1^j) + (\Delta \boldsymbol{W}_2^i - \Delta \boldsymbol{W}_2^j)\boldsymbol{W}_1 \right) \boldsymbol{x}\|.$$

**Definition 3** (Module-Switched Models). *Swapping one layer between $\mathcal{M}^i$ and $\mathcal{M}^j$ yields two possible switched models, each with its own parameters, semantic-backdoor decomposition:*

$$\theta^{ij} := \{\boldsymbol{W}_1 + \Delta \boldsymbol{W}_1^i, \, \boldsymbol{W}_2 + \Delta \boldsymbol{W}_2^j\}, \quad \mathcal{M}^{ij}(\boldsymbol{x}) = (\boldsymbol{S} + \boldsymbol{B}^{ij})\boldsymbol{x}, \quad \boldsymbol{B}^{ij} = \boldsymbol{W}_2 \Delta \boldsymbol{W}_1^i + \Delta \boldsymbol{W}_2^j \boldsymbol{W}_1,$$

$$\theta^{ji} := \{\boldsymbol{W}_1 + \Delta \boldsymbol{W}_1^j, \, \boldsymbol{W}_2 + \Delta \boldsymbol{W}_2^i\}, \quad \mathcal{M}^{ji}(\boldsymbol{x}) = (\boldsymbol{S} + \boldsymbol{B}^{ji})\boldsymbol{x}, \quad \boldsymbol{B}^{ji} = \boldsymbol{W}_2 \Delta \boldsymbol{W}_1^j + \Delta \boldsymbol{W}_2^i \boldsymbol{W}_1.$$

**Definition 4** (Distance between Outputs from Switched and Constituent Models). *Under identity activation, $\ell_2$ distances between the switched model $\mathcal{M}^{ij}$ and the two constituent models are:*

$$\|\mathcal{D}^{ij,i}\| = \|\mathcal{M}^{ij}(\boldsymbol{x}) - \mathcal{M}^i(\boldsymbol{x})\| = \|(\Delta \boldsymbol{W}_2^j - \Delta \boldsymbol{W}_2^i)\boldsymbol{W}_1 \boldsymbol{x}\|,$$

$$\|\mathcal{D}^{ij,j}\| = \|\mathcal{M}^{ij}(\boldsymbol{x}) - \mathcal{M}^j(\boldsymbol{x})\| = \|\boldsymbol{W}_2(\Delta \boldsymbol{W}_1^i - \Delta \boldsymbol{W}_1^j)\boldsymbol{x}\|.$$

*The analogous results of $\|\mathcal{D}^{ji,i}\|$ and $\|\mathcal{D}^{ji,j}\|$ hold with swapped indices (see Equation (5)).*

**Theorem 1** (Module Switching Exceeds WAG in Backdoor Divergence). *Under identity activation, the total backdoor divergence of the Weight-Averaged (WAG) model is upper bounded by the average divergence of the switched models:*

$$\|\mathcal{D}^{\mathrm{wag},i}\| + \|\mathcal{D}^{\mathrm{wag},j}\| \leq \frac{1}{2}\left(\|\mathcal{D}^{ij,i}\| + \|\mathcal{D}^{ij,j}\| + \|\mathcal{D}^{ji,i}\| + \|\mathcal{D}^{ji,j}\|\right). \tag{2}$$

This theorem confirms the rationale that module switching on average yields stronger suppression of backdoor-specific patterns than weight averaging.

**Proposition 1** (The Existence of a More Divergent Switched Model). *Given Theorem 1, there is at least one switched model with greater backdoor divergence than Weight-Averaged (WAG) model:*

$$\|\mathcal{D}^{\mathrm{wag},i}\| + \|\mathcal{D}^{\mathrm{wag},j}\| \leq \max\left\{\|\mathcal{D}^{ij,i}\| + \|\mathcal{D}^{ij,j}\|,\ \|\mathcal{D}^{ji,i}\| + \|\mathcal{D}^{ji,j}\|\right\}. \tag{3}$$

This proposition shows that the least backdoor-aligned switched model exceeds the WAG model in backdoor divergence, underscoring the importance of selecting the least aligned candidate and motivating the selection step in Section 4.4. Appendix D details proofs of Theorem 1 and Proposition 1.

**Empirical Study.** We simulate 1000 two-layer networks (with both linear and non-linear activations), each *pretrained* on a shared semantic component $\boldsymbol{S} \sim \mathcal{N}(\boldsymbol{0}, 1)$ and *fine-tuned* with a backdoor component $\boldsymbol{B}^* \sim \mathcal{N}(\boldsymbol{0}, 0.1^2)$. For each fine-tuned pair $\mathcal{M}^i$ and $\mathcal{M}^j$, we construct the corresponding WAG model $\mathcal{M}^{\mathrm{wag}}$ and switched models $\mathcal{M}^{ij}$ and $\mathcal{M}^{ji}$. We evaluate output alignment with (1) the semantic direction $\boldsymbol{Sx}$, measured by $d_S = \|\mathrm{norm}(f(\boldsymbol{x};\theta)) - \mathrm{norm}(\boldsymbol{Sx})\|$; and (2) the backdoor direction $\boldsymbol{B}^*\boldsymbol{x}$, measured by $d_B = \|\mathrm{norm}(f(\boldsymbol{x};\theta) - \boldsymbol{Sx}) - \mathrm{norm}(\boldsymbol{B}^*\boldsymbol{x})\|$, where $\mathrm{norm}(\boldsymbol{v}) = \boldsymbol{v}/\|\boldsymbol{v}\|$.

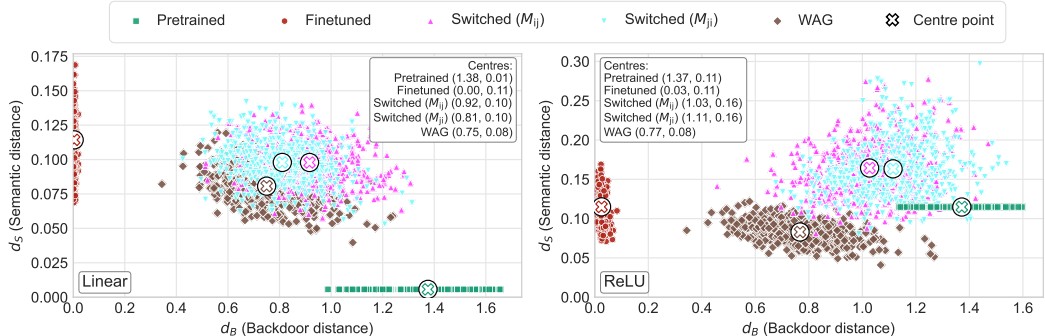

Figure 2: Euclidean distances between normalized output vectors of *pretrained*, *fine-tuned*, *WAG*, and *switched* two-layer networks, relative to the semantic direction $\boldsymbol{Sx}$ and the backdoor directions $\boldsymbol{B}^*\boldsymbol{x}$, under linear (left) and ReLU (right) activations.

Figure 2 presents 2D scatter plots comparing output distances across all model types under both linear and ReLU [1, 35] activations. More results with various activations are provided in Appendix E. We observe that while *fine-tuned* models stay close to their respective backdoor patterns $\boldsymbol{B}^*$, the *WAG* model shifts farther away, and the *switched* models diverge even more, indicating stronger backdoor suppression. All models remain near the semantic term $\boldsymbol{S}$, confirming preserved functionality.

## 4 Module Switching Defense

In this section, we extend the findings on module switching to more complicated deep neural networks and develop a comprehensive defense pipeline. We begin by introducing the problem setting in Section 4.1, followed by establishing a set of heuristic rules to guide the search for effective module switching strategies in Section 4.2. Next, we adapt an evolutionary algorithm for searching the optimal strategy in Section 4.3, guiding switched models construction and selection in Section 4.4.

### 4.1 Preliminaries

**Threat Model.** We study data poisoning attacks where an attacker modifies a subset of a clean dataset $\mathcal{D}_c = \{(x_c, y_c)\}$ into poisoned samples $\mathcal{D}_p = \{(x_p = g_t(x_c), y_p)\}$ using a trigger function $g_t$ and target label $y_p$. The poisoned data is used to train a backdoored model or shared with others for training, resulting in trojaned models being widely available via model-sharing platforms.

159 **Defender Capability.** The defender downloads potentially compromised models and aims to purify
160 them before deployment. They have white-box access and a small clean validation set (20–50 samples
161 per class), but no knowledge of the trigger or poisoned data. They can access multiple (as few as two)
162 domain-relevant models of uncertain integrity and may combine them using the validation set.

163 **Neural Network Architecture.** We adopt Transformer models [48] as the testbed in both text and
164 vision domains, given their strong performance and prevalence on model-sharing platforms. A typical
165 Transformer has $L$ layers, each with a self-attention block and a feed-forward network (FFN). The
166 attention block includes $\{W_q, W_k, W_v, W_o\}$ and the FFN includes $\{W_i, W_p\}$; we refer to these six
167 modules as $\{Q, K, V, O, I, P\}$. Residual connections [15] follow both blocks and link to later layers.

## 4.2 Scoring Rules for Module Switching

169 In Section 3, we studied weight switching in two-layer networks, where replacing weights disrupts
170 spurious correlations, eliminating undesired patterns while preserving semantic alignment. Extending
171 to DNNs, we hypothesize that breaking backdoor propagation paths can similarly deactivate them.

172 Given the structural complexity of deep networks, we define heuristic rules to guide the search for
173 module combinations that disrupt backdoor paths in both feedforward and residual streams [11]. We
174 identify three types of adjacency that may support poison transmission (illustrated in Figure 3): (1)
175 intra-layer (within the same layer), (2) consecutive-layer (adjacent layers), and (3) residual (via skip
176 connections). Additionally, we introduce a (4) balance penalty to avoid overusing any single model
177 and a (5) diversity reward to encourage varied combinations across layers.

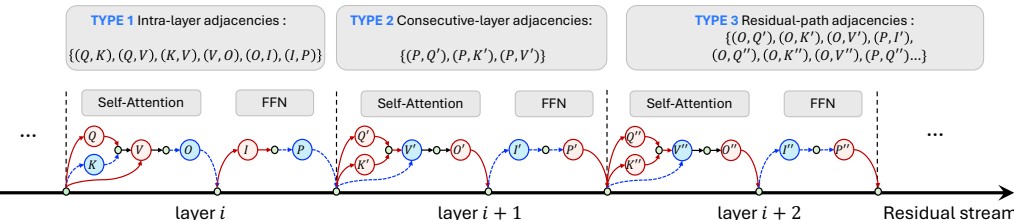

Figure 3: The confused model combines modules from different models, where red and blue nodes
indicate components from different models by considering three types of module adjacency in
Transformers, as shown in the upper part of the figure.

178 We adopt these heuristic rules as evaluation criteria to compute the overall score of a given module-
179 switching strategy, evaluating how well it adheres to the proposed principles. A summary of the rules
180 is presented in the box below (and more details are provided in Appendix F).

---

**Heuristic-based Search Rules**

1. **Intra-layer adjacency penalty:** Penalize if two adjacent modules within the same layer
   (*e.g.*, $Q$ and $K$) are from the same source model.
2. **Consecutive-layer adjacency penalty:** Penalize if adjacent modules across consecutive
   layers (*e.g.*, $P$ in layer $i$ and $Q'$ or $K'$ in layer $i+1$) are from the same source model.
3. **Residual-path adjacency penalty:** Penalize if residually connected modules (*e.g.*, $O_i \rightarrow Q', Q''$) come from the same model, with reduced weight for longer-range links.
4. **Balance penalty:** Penalize if the selected modules are skewed toward a single model.
5. **Diversity reward:** Promote layer-wise diversity, aiming at using different source model
   combinations across the network.

181

---

## 4.3 Evolutionary Module Switching Search

183 We frame the search for effective module-switching strategies as a discrete Neural Architecture
184 Search (NAS) problem [53]. Let $\mathcal{S}$ denote the space of switching strategies, where each $s \in \mathcal{S}$ assigns
185 a source model index to each module: $s : \{1, \ldots, L\} \times M \rightarrow \{1, \ldots, N\}, M = \{Q, K, V, O, I, P\}$,
186 where $L$ is the number of layers and $N$ the number of source models.

**Fitness Evaluation.**  Each strategy $s$ is scored by:

$$F(s) = -\lambda_1 A_{\text{intra}}(s) - \lambda_2 A_{\text{cons}}(s) - \lambda_3 A_{\text{res}}(s) - \lambda_4 B_{\text{bal}}(s) + \lambda_5 R_{\text{div}}(s), \qquad (4)$$

where $A_{\text{intra}}$, $A_{\text{cons}}$, and $A_{\text{res}}$ penalize adjacency violations (Section 4.2), $B_{\text{bal}}$ penalizes module imbalance, and $R_{\text{div}}$ rewards diversity. By default, we set all $\lambda_k$ to 1.0. Higher $F(s)$ indicates stronger disruption of potential backdoor paths. The formulations of all terms are provided in Appendix F.2.

**Search Algorithm.**  As the scores by $F(s)$ is non-differentiable over a large discrete space, we adopt evolutionary search [34], well suited to optimizing implicit objectives [69]. We adopt the aging regularized evolution algorithm [41], modifying it in two key ways: (1) fitness is computed directly using the heuristic scoring function $F$, without model training or validation; and (2) low-scoring strategies are discarded, replacing aging regularization [42]. As outlined in Algorithm 1, it evolves a population through tournament selection (line 11), mutation (line 12), and fitness-based dropping (line 13). A hyperparameter $C$ controls the number of children per generation. Appendix I presents example searched strategies.

### 4.4 Switched Models Construction and Selection

The searched strategy $T$ can be used to switch modules among a group of victim models $\mathcal{M} = \{\mathcal{M}_1, \ldots, \mathcal{M}_N\}$ to fuse a *candidate pool*, which on average exceeds the WAG model in backdoor divergence (as Theorem 1) and guarantees the existence of at least one candidate with higher divergence (as Proposition 1). This motivates us to develop a feature-distance-based method to identify and select the least-backdoor-aligned candidate from the pool.

**Suspect-class Detection.** We first use the final-layer embedding of [CLS] token to detect the suspect class, based on the insight that trojaned models prioritize trigger features [12, 62]. For each $m \in \mathcal{M} \cup \{\text{WAG}(\mathcal{M})\}$ and class $c$, we optimize a random input to induce prediction of $c$, yielding a dummy final-layer [CLS] feature $z_{m,c}^{\text{dum}}$. Its average cosine distance to clean features over a few non-$c$ samples is accumulated across models: $S(c) = \sum_m \text{avg}\big[1 - \cos(z_{m,c}^{\text{dum}}, z_{m,\neg c}^{\text{clean}})\big]$. The class with the highest score, $c^* = \arg\max_c S(c)$, is deemed suspicious, and the corresponding WAG dummy feature $z^* = z_{\text{WAG},c^*}^{\text{dum}}$ is used as a fixed reference.

---

**Algorithm 1** Evolutionary Module-Switching Search

1: **Input:** population $P$, generations $G$, children per generation $C$, number of models $N$, layers $L$, module set $M$.
2: $population \leftarrow \varnothing$
3: $gen\_count \leftarrow 0$
4: **while** $|population| < P$ **do**
5:     $indiv.strategy \leftarrow \textsc{RandomStrategy}(N, L, M)$
6:     $indiv.fitness \leftarrow \textsc{CalcScore}(indiv.strategy)$
7:     $population.append(indiv)$
8: **end while**
9: **while** $gen\_count < G$ **do**
10:     **for** $i \leftarrow 1$ **to** $C$ **do**
11:       $parent \leftarrow \textsc{TournamentSelect}(population)$
12:       $child.strategy \leftarrow \textsc{Mutation}(parent)$
13:       $child.fitness \leftarrow \textsc{CalcScore}(child.strategy)$
14:       $population.append(child)$
15:     **end for**
16:     $sort(population)$     ▷ by descending $fitness$ score
17:     $population \leftarrow population[0 : P]$
18:     $gen\_count \leftarrow gen\_count + 1$
19: **end while**
20: **Output:** $\textsc{BestStrategy} \leftarrow population[0].strategy$

---

**Algorithm 2** Switched Model Selection

1: **Input:** Victim models $\mathcal{M} = \{\mathcal{M}_1, \ldots, \mathcal{M}_N\}$; clean set $\mathcal{D}_c$; switching strategy $T$.
2: $wag \leftarrow \text{WAG}(\mathcal{M})$     ▷ weight averaging over $\mathcal{M}$
3: $models \leftarrow \mathcal{M} \cup \{wag\}$
4: $score \leftarrow \textsc{ZeroVector}(\texttt{num\_classes})$
5: **for** $m \in models$ **do**
6:     **for** $c \in$ candidate classes **do**
7:       $x_{\text{dummy}} \leftarrow \textsc{OptimizeInput}(m, x_{\text{random}}, c)$
8:       $z_{\text{dummy}} \leftarrow \textsc{Forward}(m, x_{\text{dummy}})$
9:       $z_{\text{clean}} \leftarrow \textsc{Forward}(m, \mathcal{D}_c, \texttt{non-}c)$
10:       $score[c] \mathrel{+}= \textsc{MeanCosineDist}(z_{\text{dummy}}, z_{\text{clean}})$
11:       $\textsc{DummyFeature}[m][c] \leftarrow z_{\text{dummy}}$
12:     **end for**
13: **end for**
14: $c^* \leftarrow \arg\max_c score[c]$     ▷ suspect target class
15: $z^* \leftarrow \textsc{DummyFeature}[wag][c^*]$
16: $candidates \leftarrow \textsc{ModuleSwitch}(T, \mathcal{M})$
17: **for** $m \in candidates$ **do**
18:     $z \leftarrow \textsc{Forward}(m, \mathcal{D}_c, \texttt{non-}c^*)$
19:     $m.dist \leftarrow \textsc{MeanCosineDist}(z, z^*)$
20: **end for**
21: **Output:** $\arg\max_m m.dist$

---

**Candidate Selection.** Applying $T$ to $\mathcal{M}$ gives candidates $m \in \mathcal{C}(T, \mathcal{M})$ (*e.g.*, $\mathcal{M}^{ij}, \mathcal{M}^{ji}$). Each $m$ is scored by $d(m) = \text{avg}\left[1 - \cos(z^*, f_m(\boldsymbol{x}))\right]$, the mean cosine distance between its [CLS] features on a few clean, non-$c^*$ samples $\boldsymbol{x}$ and the WAG dummy $z^*$. The winner $m^* = \arg\max_{m \in \mathcal{C}(T, \mathcal{M})} d(m)$ is the one least aligned with backdoor features and, by Proposition 1, has better defense than WAG.

The complete pipeline, detailed in Algorithm 2, avoids exhaustive trojan detection process [31, 44, 49, 50, 52], yet reliably selects robust module-switching candidates.

## 5 Experiments

### 5.1 Experimental Setup

**Datasets.** We evaluate our method on three NLP datasets–**SST-2** [24, 43], **MNLI** [54], and **AG News** [66]–as well as vision datasets, **CIFAR-10** [21, 46] and **TinyImageNet** [23], covering both binary and multi-class classification. Dataset statistics are in Table 6 (Appendix G.1). For NLP, following WAG [2], we use 20% poison in training (also testing 10% and 1%). For vision tasks, we apply a 5% poison rate. Poisoned test sets are created by attacking non-target validation samples; only the clean test set is available to the defender, while poisoned test data is used solely for evaluation.

**Backdoor Attacks.** We generate poisoned data by modifying clean samples and relabelling them to a target class, using four representative attacks in both text and vision tasks, to evaluate our defense.

For the text domain, we consider (1) **BadNet** [22], (2) **InsertSent** [7], (3) Learnable Word Substitution (**LWS**) [40], and (4) Hidden-Killer (**Hidden**) [38]. **BadNet** and **InsertSent** are token and sentence insertion attacks, and we set the triggers as rare words {*"cf"*, *"mn"*, *"bb"*, *"tq"*, *"mb"*} and phrases {*"I watched this movie"*, *"no cross, no crown"*}. **LWS** and **Hidden** apply stealthier strategies such as synonym substitution and syntactic paraphrasing.

For the vision domain, we examine (1) **BadNet** [14], (2) **WaNet** [36], (3) **BATT** [58], and (4) **PhysicalBA** [27]. BadNet and BATT inject digital patterns such as fixed pixel triggers and subtle visual changes, while PhysicalBA and WaNet are stealthier and use physical objects and warping effects. We utilize the BackdoorBox [28] toolkit to generate poisoned datasets and train the models.

**Defense Baselines.** We compare against seven defense methods across text and vision: three model-merging approaches applicable to both domains–**TIES** [59], **DARE** [63], and **WAG** [2]–and two domain–specific data purification methods per modality. **Z-Def.** [17] and **ONION** [37] are outlier detection methods in text domain. In vision, **CutMix** [64] disrupts triggers via patch mixing, and **ShrinkPad** [27] reduces vulnerability by shrinking and padding inputs. All baselines use open-source implementations with default settings. See Appendix G.3 for more details.

**Evaluation Metrics.** We assess the model's utility and defense performance using Clean Accuracy (**CACC**) and Attack Success Rate (**ASR**) [2, 17, 37, 40]. CACC measures the prediction accuracy on clean samples, with a higher CACC indicating better model utility. ASR computes the attack accuracy on a poisoned test set, where all test samples are attacked and their labels are modified to the target class. A higher ASR reflects that the model is more vulnerable to the attack.

**Implementation Details.** We use *RoBERTa-large* [32], *BERT-large* [8], and *DeBERTa-large* [16] for text experiments, and *Visual transformers (ViT)* [55] for vision tasks. NLP models are fine-tuned on poisoned data for 3 epochs using Adam [20] with a learning rate of $2 \times 10^{-5}$; ViT models for 10 epochs using SGD [3] at $1 \times 10^{-2}$. We focus on two-model merging in both domains and include three-model merging for text. All experiments are run with three random seeds on a single Nvidia A100 GPU, reporting average results. The evolutionary search runs for 2 million generations on a single CPU (6 hours for the setup with 24 layers times 6 modules per layer). As the strategy is structure-driven and task-agnostic, it only requires single searched per architecture. For model selection discussed in Section 4.4, we use 50 samples per class as the evaluation set for selecting candidate models, and we further ablate the quantity to 20 samples per class in Section 5.3.

### 5.2 Main Results

**Mitigation of Textual Backdoor Attacks.** We evaluate our defense method using *RoBERTa-large* on three datasets: **SST-2**, **MNLI**, and **AG News**. Partial results for SST-2 are shown in Table 1, with full results in Appendix H.1. We consider two types of two-model combinations: (1) six

Table 1: Performance comparison across backdoor attacks on **SST-2** using *RoBERTa-large*. Best results are in blue . * indicates results averaged over four variants; same for subsequent tables.

| Defense | CACC | Attack Success Rate (ASR)↓ | | | | | Defense | CACC | Attack Success Rate (ASR)↓ | | | | |
|---|---|---|---|---|---|---|---|---|---|---|---|---|---|
| | | BadNet | Insert | LWS | Hidden | AVG. | | | BadNet | Insert | LWS | Hidden | AVG. |
| Benign | 95.9 | 4.1 | 2.2 | 12.8 | 16.5 | 8.9 | Z-Def | 95.6* | 4.6 | 1.8 | 97.3 | 35.7 | 34.9 |
| Victim | 95.9* | 100.0 | 100.0 | 98.0 | 96.5 | 98.6 | ONION | 92.8* | 56.8 | 99.9 | 85.7 | 92.9 | 83.8 |
| | | *Combined: BadNet + InsertSent* | | | | | | | *Combined: BadNet + HiddenKiller* | | | | |
| WAG | 96.3 | 56.3 | 7.4 | - | - | 31.9 | WAG | 96.1 | 63.9 | - | - | 29.0 | 46.4 |
| TIES | 95.9 | 88.7 | 17.0 | - | - | 52.9 | TIES | 96.0 | 90.4 | - | - | 36.9 | 63.6 |
| DARE | 96.5 | 57.8 | 36.3 | - | - | 47.1 | DARE | 96.7 | 36.5 | - | - | 47.6 | 41.9 |
| Ours | 96.2 | 36.9 | 7.1 | - | - | 22.0 | Ours | 96.1 | 40.5 | - | - | 27.7 | 34.1 |
| | | *Combined: BadNet + LWS* | | | | | | | *Combined: Benign + BadNet* | | | | |
| WAG | 96.2 | 74.0 | - | 50.3 | - | 62.2 | WAG | 96.1 | 39.3 | - | - | - | 39.3 |
| TIES | 95.9 | 88.1 | - | 66.1 | - | 77.1 | TIES | 95.7 | 69.2 | - | - | - | 69.2 |
| DARE | 96.2 | 60.4 | - | 62.5 | - | 61.4 | DARE | 96.4 | 43.2 | - | - | - | 43.2 |
| Ours | 96.0 | 41.7 | - | 39.0 | - | 40.4 | Ours | 96.1 | 12.2 | - | - | - | 12.2 |

Table 2: Performance comparison across backdoor attacks on the **CIFAR-10** dataset using *ViT*.

| Defense | CACC | BadNet | WaNet | BATT | PBA | AVG. | Defense | CACC | BadNet | WaNet | BATT | PBA | AVG. |
|---|---|---|---|---|---|---|---|---|---|---|---|---|---|
| Benign | 98.8 | 10.1 | 10.2 | 7.7 | 10.1 | 9.5 | CutMix | 97.7* | 87.1 | 70.6 | 99.9 | 64.9 | 80.6 |
| Victim | 98.5* | 96.3 | 84.7 | 99.9 | 89.4 | 92.6 | ShrinkPad | 97.3* | 14.4 | 51.3 | 99.9 | 88.3 | 63.5 |
| | | *Combined: BadNet + WaNet* | | | | | | | *Combined: BadNet + BATT* | | | | |
| WAG | 98.7 | 13.7 | 10.6 | - | - | 12.2 | WAG | 98.9 | 10.1 | - | 42.9 | - | 26.5 |
| TIES | 98.6 | 11.9 | 10.7 | - | - | 11.3 | TIES | 98.9 | 10.1 | - | 47.9 | - | 29.0 |
| DARE | 98.8 | 83.3 | 10.2 | - | - | 46.7 | DARE | 99.0 | 69.2 | - | 26.8 | - | 48.0 |
| Ours | 98.7 | 12.3 | 10.5 | - | - | 11.4 | Ours | 98.7 | 10.2 | - | 32.6 | - | 21.4 |
| | | *Combined: BadNet + PhysicalBA* | | | | | | | *Combined: Benign + PhysicalBA* | | | | |
| WAG | 99.0 | 39.6 | - | - | 39.5 | 39.6 | WAG | 99.0 | - | - | - | 10.1 | 10.1 |
| TIES | 99.0 | 38.9 | - | - | 38.9 | 38.9 | TIES | 98.8 | - | - | - | 10.2 | 10.2 |
| DARE | 99.0 | 72.2 | - | - | 72.2 | 72.2 | DARE | 99.9 | - | - | - | 10.1 | 10.1 |
| Ours | 98.7 | 18.5 | - | - | 18.4 | 18.5 | Ours | 98.9 | - | - | - | 10.1 | 10.1 |

pairwise merges of four backdoored models, and (2) four cases where a benign model is combined with backdoored ones to evaluate unintended vulnerability exposure. We employ a unified strategy obtained via our evolutionary algorithm (see Figure 6) and apply it consistently across all settings.

Across all three datasets and different model pairs, our method consistently achieves strong defense performance compared to baselines while maintaining high clean accuracy scores. For example, when combining models with two insertion-based attacks BadNet and InsertSent, our method reduces the average ASR to 22.0%, compared to 31.9% for the best baseline WAG. When combining BadNet with LWS (a more stealthy attack), our method achieves an ASR of 40.4%, providing at least a 21.0% absolute improvement over baselines (typically above 60%). This shows that even when merging compromised models, our method effectively disrupts spurious correlations and defends against backdoor attacks.

When merging a benign model with compromised ones, our method achieves a low ASR across four combinations, with the BadNet-controlled group achieving 12.2%, which is 27.1% better than the best baseline WAG. This suggests that our method effectively prevents unintended backdoor effects, unlike other approaches that prioritize downstream utility but inadvertently introduce such vulnerabilities. Additionally, while the baseline Z-Def demonstrates strong effectiveness against the insertion-based attacks BadNet and InsertSent (with access to training data), it is less effective at defending against the LWS and HiddenKiller attacks due to their subtle trigger pattern design.

**Mitigation of Vision Backdoor Attacks.** We assess our method on the **CIFAR-10** and **TinyImageNet** datasets using a 12-layer *ViT* [55] model. Partial results for CIFAR-10 are shown in Table 2, with full results presented in Appendix H.2. The evolutionary search yields the module-switching strategy in Figure 12, applied across all vision experiments.

Our method consistently defends against all attack combinations while preserving utility. For example, in the BadNet + PhysicalBA case, it lowers ASR to 18.5%, outperforming all baselines by at least 20.4%. These results demonstrate the robustness of our strategy in disrupting spurious correlations and its effectiveness across domains with different input characteristics.

**Three-Model Fusion Defense.** We further evaluate our method fusing three models tested in the text domain, applying the strategy shown in Figure 11. The results are presented in Table 3.

Table 3: Results of combining three backdoored models on **SST-2**. Best results are **highlighted**.

| Defense | CACC | BadNet | Insert | LWS | Hidden | AVG. | Defense | CACC | BadNet | Insert | LWS | Hidden | AVG. |
|---|---|---|---|---|---|---|---|---|---|---|---|---|---|
| *BadNet + InsertSent + LWS* | | | | | | | *BadNet + InsertSent + HiddenKiller* | | | | | | |
| WAG | 96.3 | 9.5 | **3.4** | **21.6** | - | **11.5** | WAG | 96.7 | 5.9 | 2.7 | - | 19.1 | 9.2 |
| Ours | 96.0 | **9.2** | 3.8 | 25.9 | - | 13.0 | Ours | 96.2 | 5.9 | **1.6** | - | **18.7** | **8.7** |
| *BadNet + LWS + HiddenKiller* | | | | | | | *InsertSent + LWS + HiddenKiller* | | | | | | |
| WAG | 96.0 | 10.8 | - | 30.9 | **20.3** | 20.7 | WAG | 96.0 | - | 2.7 | 25.5 | 19.6 | 15.9 |
| Ours | 96.2 | **7.9** | - | **25.7** | 20.7 | **18.1** | Ours | 96.2 | - | **2.1** | **24.1** | **19.4** | **15.2** |

Among the four possible combinations from our victim model pool, our method identified the optimal configuration in three cases. Even in the remaining case (BadNet, InsertSent, and LWS), the defense remained strong, achieving a low average ASR of 13.0%. For the optimal combinations, our method consistently outperformed a strong baseline with ASRs already below 20%, demonstrating improved defense effectiveness. These results highlight our approach's ability to disrupt multiple spurious correlations and maintain robustness in multi-model fusion.

**Comparison of Different Strategies.** We compare two evolutionary search strategies–with and without early stopping–shown in Figures 6 and 7, and report their fitness scores in Table 12 of Appendix H.3. The early stopping terminates the search when no improvement in fitness score is observed over 100,000 iterations. We observe a positive correlation between the fitness score and defense performance: the adopted strategy without early stopping achieves a higher score and reduces the ASR by 27.2%. Based on score breakdowns and visualizations, we attribute the improvement to fewer residual rule violations, which more effectively disrupt subtle spurious correlations.

**Candidate Selection Results.** Our method generates multiple asymmetric module allocation candidates, with selection guided by the process in Section 4.4. While the selected candidate consistently performs well, we also analyze the unselected ones (see Table 13 in Appendix H.4). In most cases, our method correctly identifies the top-performing candidate, outperforming other options by a significant margin. Even when an unselected candidate achieves a lower ASR in specific cases, our chosen candidate remains competitive with both the best alternative and the WAG baseline.

## 5.3 Ablation Studies

**Importance of Heuristic Rules.** We ablate each of the first three rules from Section 4.2 to evaluate their individual contributions. As shown in Table 14 (Appendix H.5), removing any rule typically degrades performance, highlighting the complementary effect of the full rule set. Visualizations in Figures 8 to 10 show that each ablation yields distinct strategy patterns.

**Generalization across Architectures.** We apply our method to *RoBERTa-large*, *BERT-large*, and *DeBERTa-v3-large* under three settings. As shown in Table 15 (Appendix H.6), our approach consistently outperforms WAG across all tests. Importantly, we reuse the same searched strategy from Figure 6, demonstrating strong cross-model generalization and supporting practical scalability.

**Minimum Clean Data Requirement.** We examine the impact of reducing clean supervision from 50 to 20 samples per class on SST-2 across three architectures. Results in Table 15 (Appendix H.7) show our method still selects low-ASR candidates, suggesting effectiveness with limited clean data.

**Performance under Varying Poisoning Rates.** We test robustness under 20%, 10%, and 1% poisoning rates on SST-2 using *RoBERTa-large*. As shown in Table 16 (Appendix H.8), our method consistently achieves lower ASR than WAG across different attacks and poisoning levels.

## 6 Conclusion

In this paper, we propose Module-Switching Defense (MSD), a post-training backdoor defense that disrupts shortcuts of spurious correlations by strategically switching weight modules between (compromised) models. MSD does not rely on trusted reference models or training data and remains effective with a couple of models. Using heuristic rules and evolutionary search, we establish a transferable module confusion strategy that mitigates various backdoor attacks while preserving their task utility. Empirical results on text and vision tasks confirm its outstanding defense performance, and strong generalization capability, highlighting its practicality in real-world applications.

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

## A  Limitations

While our study demonstrates the effectiveness of Module-Switching Defense (MSD) across a range of classification tasks in NLP and CV, we identify two main limitations. First, our focus is restricted to classification-based settings. Backdoor attacks in generative models operate through notably different mechanisms, and extending MSD to such scenarios remains an important direction for future research. Second, our method is designed and evaluated primarily within Transformer-based architectures, which dominate current text and vision benchmarks. The applicability of MSD to other model families, such as convolutional neural networks (CNNs) or emerging architectures, is left for future exploration.

## B  Broader Impacts

This paper presents an efficient post-training defense against backdoor attacks on deep neural networks. By strategically combining model weight modules from either clean or compromised models, our approach disrupts backdoor propagation while preserving model utility. We demonstrated the usage of MSD to strengthen the security of machine learning models in both natural language processing and computer vision. All models and datasets used in this study are sourced from established open-source platforms. The discovered MSD templates will be released to facilitate further research on defense study. While we do not anticipate any direct negative societal consequences, we hope this work encourages further research into more robust defense mechanisms.

## C  Empirical validation of the second-order interaction magnitude

We empirically validate the condition adopted in Section 3, where the second-order interaction term $\epsilon = \Delta W_2 \Delta W_1$ is omitted due to its negligible magnitude relative to the first-order terms. This validation proceeds from three perspectives.

First, Figure 4 compares the Frobenius norms of the semantic term $S = W_2 W_1$, the first-order adaptation term $B = W_2 \Delta W_1 + \Delta W_2 W_1$, and the second-order residual $\epsilon = \Delta W_2 \Delta W_1$ across five derived networks. The left subfigure confirms that $\|\epsilon\|$ is consistently two orders of magnitude smaller than $\|S\|$ and well below $4\%$ of $\|B\|$. The right subfigure further reveals that the element-wise values of $\epsilon$ concentrate tightly around zero, contrasting with the heavier tails of $B$ and $S$.

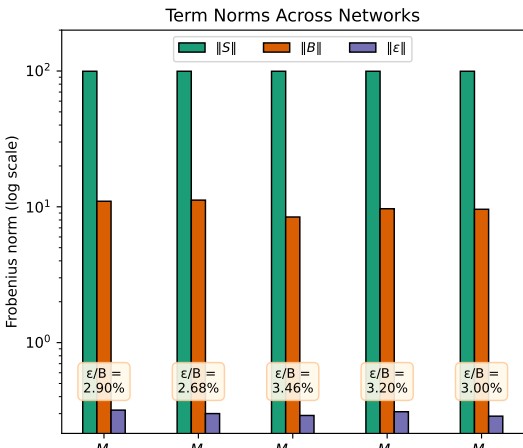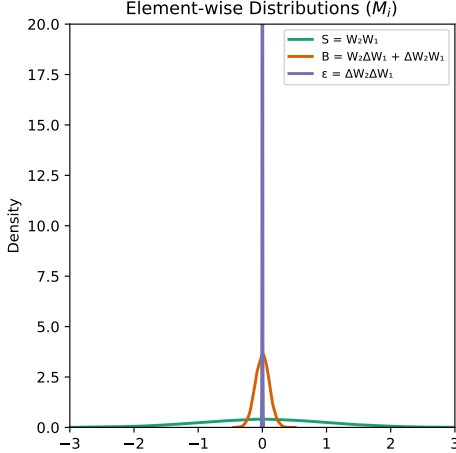

Figure 4: Frobenius norm and element-wise distribution of the semantic, first-order, and second-order terms across five network configurations. While the first-order term dominates the residual behavior, the second-order interaction $\epsilon = \Delta W_2 \Delta W_1$ remains negligible in both scale and distribution.

Second, Table 4 reports $\|\epsilon\|/\|B\|$ ratios across five network variants under varying backdoor strengths, where perturbations are sampled from zero-mean Gaussian noise with increasing variance. The inclusion of error bars (mean $\pm$ standard deviation) reflects variation across multiple runs. In typical scenarios where the backdoor signal is weak or comparable to the main semantic component,

the second-order interaction consistently remains below $4\%$ of the first-order term. Even under exaggerated settings where the backdoor signal is scaled to $1.5\times$ or $2\times$ the semantic strength, $\|\epsilon\|/\|\boldsymbol{B}\|$ remains within a stable range of $5\%$–$7\%$, reaffirming the negligible and bounded nature of second-order interactions across regimes.

Table 4: Relative magnitude of second-order interactions, reported as $\|\varepsilon\|/\|\boldsymbol{B}\|$, across networks and backdoor strengths. All models are evaluated with $\boldsymbol{S} \sim \mathcal{N}(\boldsymbol{0}, 1)$ and perturbations $\boldsymbol{B} \sim \mathcal{N}(\boldsymbol{0}, \sigma^2)$.

| Semantic Dist | Backdoor Dist | $\|\varepsilon\|/\|\boldsymbol{B}\|$ for Different Shallow Models | | | | |
| --- | --- | --- | --- | --- | --- | --- |
| | | $\mathcal{M}^i$ | $\mathcal{M}^j$ | $\mathcal{M}^{wag}$ | $\mathcal{M}^{ij}$ | $\mathcal{M}^{ji}$ |
| | $\boldsymbol{B} \sim \mathcal{N}(\boldsymbol{0}, 0.1^2)$ | 2.82±0.19% | 2.70±0.20% | 3.42±0.23% | 2.95±0.15% | 3.14±0.21% |
| | $\boldsymbol{B} \sim \mathcal{N}(\boldsymbol{0}, 0.5^2)$ | 1.98±0.21% | 1.90±0.28% | 1.78±0.18% | 1.75±0.12% | 1.76±0.18% |
| $\boldsymbol{S} \sim \mathcal{N}(\boldsymbol{0}, 1.0^2)$ | $\boldsymbol{B} \sim \mathcal{N}(\boldsymbol{0}, 1.0^2)$ | 3.24±0.22% | 3.10±0.32% | 2.33±0.17% | 2.33±0.12% | 2.30±0.17% |
| | $\boldsymbol{B} \sim \mathcal{N}(\boldsymbol{0}, 1.5^2)$ | 4.77±0.25% | 4.48±0.33% | 3.18±0.14% | 3.09±0.15% | 2.97±0.12% |
| | $\boldsymbol{B} \sim \mathcal{N}(\boldsymbol{0}, 2.0^2)$ | 6.31±0.27% | 6.28±0.35% | 4.14±0.29% | 4.06±0.14% | 3.92±0.18% |

Additionally, we extend this analysis to deep transformer-based [48] models by computing $\|\epsilon\|/\|\boldsymbol{B}\|$ for the attention weight product, where $\boldsymbol{W}_1$ and $\boldsymbol{W}_2$ denote the key ($K$) and query ($Q$) projection matrices, respectively, and $QK^\top := \boldsymbol{W}_2\boldsymbol{W}_1$. The weight changes $\Delta\boldsymbol{W}_1$, $\Delta\boldsymbol{W}_2$ are computed relative to the original pretrained *RoBERTa-large* [32] weights. All models are trained on **SST-2** [43], including both benign and backdoored variants such as **BadNet** [22], **InsertSent** [7], learnable word substitution (**LWS**) [40], and Hidden-Killer (**Hidden**) [38].

As shown in Table 5, across all pairwise combinations of these models, the relative magnitude of second-order interactions consistently remains below $4\%$. Each reported value reflects the mean and standard deviation computed across all 24 layers of *RoBERTa-large*. This pattern holds across both original and recombined variants ($\mathcal{M}^{\text{wag}}$, $\mathcal{M}^{ij}$, $\mathcal{M}^{ji}$), confirming the stability of second-order contributions in practical transformer settings.

Table 5: Relative magnitude of second-order interactions, reported as $\|\varepsilon\|/\|\boldsymbol{B}\|$, computed from the key ($K$) and query ($Q$) projection matrices in *RoBERTa-large* models trained on **SST-2**.

| Combination | $\|\varepsilon\|/\|\boldsymbol{B}\|$ for Attention Weight Product ($QK^T$) in *RoBERTa-large* Models | | | | |
| --- | --- | --- | --- | --- | --- |
| ($\mathcal{M}^i + \mathcal{M}^j$) | $\mathcal{M}^i$ | $\mathcal{M}^j$ | $\mathcal{M}^{wag}$ | $\mathcal{M}^{ij}$ | $\mathcal{M}^{ji}$ |
| *BadNet + InsertSent* | 3.53±0.77% | 3.24±0.61% | 2.43±0.39% | 2.68±0.51% | 2.61±0.50% |
| *BadNet + LWS* | 3.53±0.77% | 3.30±0.65% | 2.46±0.4% | 2.71±0.46% | 2.68±0.49% |
| *BadNet + Hidden* | 3.53±0.77% | 3.30±0.61% | 2.49±0.43% | 2.77±0.45% | 2.72±0.45% |
| *BadNet + Benign* | 3.53±0.77% | 3.27±0.58% | 2.52±0.42% | 2.78±0.47% | 2.73±0.48% |

Accordingly, we omit the second-order term $\epsilon$ in our definitions and proofs throughout the paper without loss of generality.

 # D   Proofs of Theorem 1 and Proposition 1

670 **Theorem 1** (Module Switching Exceeds WAG in Backdoor Divergence). *Under identity activation,*
671 *the total backdoor divergence of the Weight-Averaged (WAG) model is upper bounded by the average*
672 *divergence of the switched models:*

$$\|\mathcal{D}^{\mathrm{wag},i}\| + \|\mathcal{D}^{\mathrm{wag},j}\| \le \frac{1}{2}\left(\|\mathcal{D}^{ij,i}\| + \|\mathcal{D}^{ij,j}\| + \|\mathcal{D}^{ji,i}\| + \|\mathcal{D}^{ji,j}\|\right). \tag{2}$$

673 **Proposition 1** (The Existence of a More Divergent Switched Model). *Given Theorem 1, there is at*
674 *least one switched model with greater backdoor divergence than Weight-Averaged (WAG) model:*

$$\|\mathcal{D}^{\mathrm{wag},i}\| + \|\mathcal{D}^{\mathrm{wag},j}\| \le \max\left\{\|\mathcal{D}^{ij,i}\| + \|\mathcal{D}^{ij,j}\|,\ \|\mathcal{D}^{ji,i}\| + \|\mathcal{D}^{ji,j}\|\right\}. \tag{3}$$

675 *Proof.* From Definition 2 and 4, we have the following expressions for the backdoor divergences:

$$
\begin{aligned}
\|\mathcal{D}^{\mathrm{wag},i}\| &= \frac{1}{2}\left\|\left(\boldsymbol{W}_2(\Delta\boldsymbol{W}_1^j - \Delta\boldsymbol{W}_1^i) + (\Delta\boldsymbol{W}_2^j - \Delta\boldsymbol{W}_2^i)\boldsymbol{W}_1\right)\boldsymbol{x}\right\|, \\
\|\mathcal{D}^{\mathrm{wag},j}\| &= \frac{1}{2}\left\|\left(\boldsymbol{W}_2(\Delta\boldsymbol{W}_1^i - \Delta\boldsymbol{W}_1^j) + (\Delta\boldsymbol{W}_2^i - \Delta\boldsymbol{W}_2^j)\boldsymbol{W}_1\right)\boldsymbol{x}\right\|, \\
\|\mathcal{D}^{ij,i}\| &= \left\|(\Delta\boldsymbol{W}_2^j - \Delta\boldsymbol{W}_2^i)\boldsymbol{W}_1\boldsymbol{x}\right\|, \quad \|\mathcal{D}^{ij,j}\| = \left\|\boldsymbol{W}_2(\Delta\boldsymbol{W}_1^i - \Delta\boldsymbol{W}_1^j)\boldsymbol{x}\right\|, \\
\|\mathcal{D}^{ji,i}\| &= \left\|\boldsymbol{W}_2(\Delta\boldsymbol{W}_1^j - \Delta\boldsymbol{W}_1^i)\boldsymbol{x}\right\|, \quad \|\mathcal{D}^{ji,j}\| = \left\|(\Delta\boldsymbol{W}_2^i - \Delta\boldsymbol{W}_2^j)\boldsymbol{W}_1\boldsymbol{x}\right\|.
\end{aligned}
\tag{5}
$$

676 **Linear relationships.**   By regrouping terms in the above definitions, we obtain the following vector
677 identities:

$$\mathcal{D}^{\mathrm{wag},i} = \frac{1}{2}(\mathcal{D}^{ij,i} + \mathcal{D}^{ji,i}), \qquad \mathcal{D}^{\mathrm{wag},j} = \frac{1}{2}(\mathcal{D}^{ij,j} + \mathcal{D}^{ji,j}). \tag{6}$$

678 **Bounding the average switched model backdoor divergence.**   Substituting equation 6 into the
679 norms and applying the triangle inequality [47], we have:

680
$$\|\mathcal{D}^{\mathrm{wag},i}\| = \|\frac{1}{2}(\mathcal{D}^{ij,i} + \mathcal{D}^{ji,i})\| \le \frac{1}{2}\left(\|\mathcal{D}^{ij,i}\| + \|\mathcal{D}^{ji,i}\|\right), \tag{7}$$

$$\|\mathcal{D}^{\mathrm{wag},j}\| = \|\frac{1}{2}(\mathcal{D}^{ij,j} + \mathcal{D}^{ji,j})\| \le \frac{1}{2}\left(\|\mathcal{D}^{ij,j}\| + \|\mathcal{D}^{ji,j}\|\right). \tag{8}$$

681 Summing both inequalities gives:

$$\|\mathcal{D}^{\mathrm{wag},i}\| + \|\mathcal{D}^{\mathrm{wag},j}\| \le \frac{1}{2}\left(\|\mathcal{D}^{ij,i}\| + \|\mathcal{D}^{ji,i}\| + \|\mathcal{D}^{ij,j}\| + \|\mathcal{D}^{ji,j}\|\right), \tag{9}$$

682 which proves Theorem 1.

683 **Bounding the maximum switched model backdoor divergence.**   Let:

$$C_1 := \|\mathcal{D}^{ij,i}\| + \|\mathcal{D}^{ij,j}\|, \qquad C_2 := \|\mathcal{D}^{ji,i}\| + \|\mathcal{D}^{ji,j}\|, \qquad G := \max\{C_1, C_2\}. \tag{10}$$

684 Since $C_1 + C_2 \le 2G$, it follows that:

$$\|\mathcal{D}^{\mathrm{wag},i}\| + \|\mathcal{D}^{\mathrm{wag},j}\| \le \frac{1}{2}(C_1 + C_2) \le \max\{C_1, C_2\}, \tag{11}$$

685 which proves Proposition 1. □

# E   Module Switching with Additional Activation Functions

687 We extend the experiments from Section 3 to two additional activation functions: *tanh* and *sigmoid* [9],
688 in addition to the *linear* and *ReLU* results discussed in the main text. For each activation, we
689 simulate 1000 pairs of *fine-tuned* models $\mathcal{M}^i$ and $\mathcal{M}^j$ with a shared pretrained semantic component
690 $\boldsymbol{S} \sim \mathcal{N}(\boldsymbol{0}, 1^2)$ and individual backdoor shifts $\boldsymbol{B}^* \sim \mathcal{N}(\boldsymbol{0}, 0.1^2)$. We then construct the weight-
691 averaged model $\mathcal{M}^{\mathrm{wag}}$ and the module-switched models $\mathcal{M}^{ij}$ and $\mathcal{M}^{ji}$, as defined in Definitions 1
692 and 3.

693 Figure 5 visualizes the semantic and backdoor alignment of each model type across the four activation
694 functions. Consistently across activations, we observe that:

- *Fine-tuned* models remain closely aligned with their respective backdoor direction $B^*x$;

- *WAG* models deviate more from the backdoor pattern;

- *Switched* models exhibit the larger distance to backdoor patterns, indicating stronger mitigation;

- All model types maintain proximity to the semantic output $Sx$, confirming that semantic information is preserved.

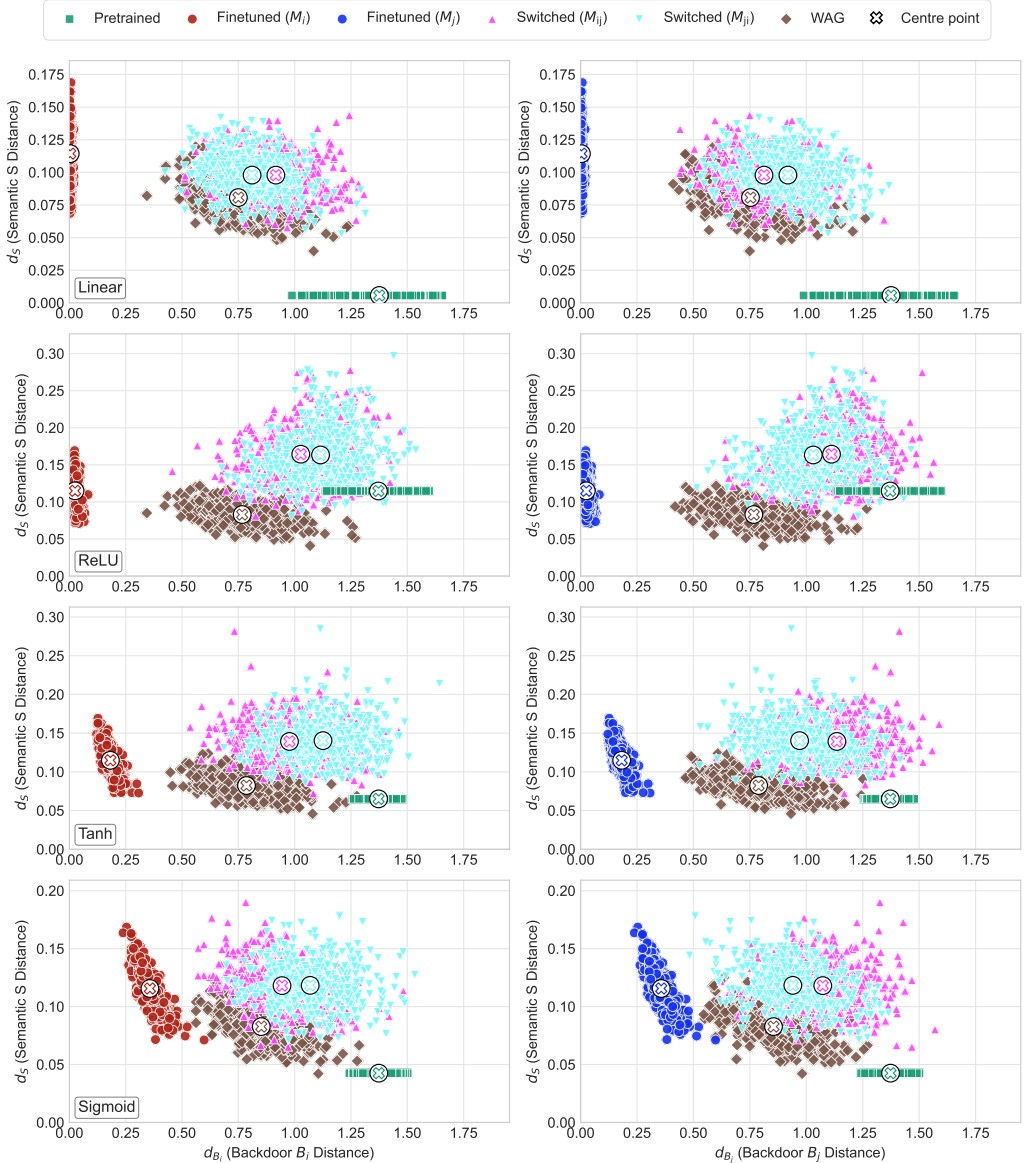

Figure 5: Euclidean distances between normalized output vectors of *pretrained*, *fine-tuned*, *WAG*, and *switched* networks, relative to semantic output $Sx$ and backdoor output $B^*x$, under *linear*, *ReLU*, *tanh*, and *sigmoid* activations.

These results generalize the findings in Figure 2 to a broader range of nonlinear activations, reinforcing the conclusion that module switching more effectively disrupts backdoor behavior while retaining semantic utility.

# F  Fitness Score Calculation for Evolutionary Search

Building upon the heuristic rules established in Section 4.2 for disrupting backdoor connections in compromised models, we develop a comprehensive fitness function. This function incorporates five key components that collectively evaluate the quality of a module composition strategy.

## F.1  Heuristic Rules

Our fitness function implements the following rules through penalties and rewards:

- **Intra-layer adjacency penalty**: Penalizes adjacent modules from the same source model within a specific layer $i$ (*e.g.*, $Q_i$ and $K_i$).

- **Consecutive-layer adjacency penalty**: Discourages direct connections between modules from the same source model across consecutive layers $i$ and $i+1$ (*e.g.*, $P_i$ to $Q_{i+1}$).

- **Residual-path adjacency penalty**: Applies a distance-weighted penalty to modules from the same source model connected via residual connections between layers $i$ and $j$ (*e.g.*, $O_i$ to $Q_j$, where $j > i$), with diminishing impact as $j - i$ increases.

- **Balance penalty**: Promotes uniform distribution of modules $\{Q, K, V, O, I, P\}$ across source models to prevent any single model from dominating the architecture.

- **Diversity reward**: Encourages varied module combinations across layers to enhance architectural diversity.

## F.2  Mathematical Formulation

As introduced in Section 4.3, the total fitness score for a given module composition strategy $s$ is:

$$F(s) = -\lambda_1 A_{\text{intra}}(s) - \lambda_2 A_{\text{cons}}(s) - \lambda_3 A_{\text{res}}(s) - \lambda_4 B_{\text{bal}}(s) + \lambda_5 R_{\text{div}}(s), \tag{12}$$

where all $\lambda_k$ are weight factors (default to 1.0) that control the relative importance of each component in the overall fitness score.

Each component is calculated as follows:

**1. Intra-layer Adjacency ($A_{\textbf{intra}}(s)$)**

$$A_{\text{intra}}(s) = -\sum_{l=1}^{|s|} \text{INTRAVIOLATION}(s[l]) \tag{13}$$

Here, INTRAVIOLATION quantifies the number of adjacent module pairs from the same source model within layer $s[l]$.

**2. Consecutive-layer Adjacency ($A_{\textbf{cons}}(s)$)**

$$A_{\text{cons}}(s) = -\sum_{l=1}^{|s|-1} \text{CONSECVIOLATION}(s[l], s[l+1]) \tag{14}$$

The function CONSECVIOLATION counts module pairs from the same source model that are directly connected between consecutive layers.

**3. Residual Connections ($A_{\textbf{res}}(s)$)**

$$A_{\text{res}}(s) = -\sum_{l=1}^{|s|} \sum_{k=l+1}^{|s|} \text{RESIDUALVIOLATION}(s[l], s[k]) \times (0.5)^{k-l} \tag{15}$$

This term evaluates residual connections between layers $s[l]$ and $s[k]$, with RESIDUALVIOLATION weighted by $(0.5)^{k-l}$ to reduce the impact of long-range connections.

**4. Module Balance ($B_{\mathbf{bal}}(s)$)**

$$B_{\text{bal}}(s) = -\sum_{i=1}^{n_{\text{models}}} \sum_{m \in \mathcal{M}} |\text{count}_{i,m} - \text{count}_{\text{ideal}}| \tag{16}$$

where $\text{count}_{i,m}$ is the count of module type $m$ from model $i$, $M = \{Q, K, V, O, I, P\}$ is the set of module types, and $\text{count}_{\text{ideal}} = |s|/n_{\text{models}}$ represents the ideal count per module type per model.

**5. Layer Diversity ($R_{\mathbf{div}}(s)$)**

$$R_{\text{div}}(s) = |\text{unique}(s)| \tag{17}$$

where $\text{unique}(s)$ is the set of unique layer compositions in strategy $s$.

## G  Additional Experiment Setup

### G.1  Dataset Statistics

We evaluate our method on four text and two vision datasets. The statistics of each dataset and the settings of backdoor target class are shown in Table 6.

Table 6: The statistics of the evaluated text and vision datasets.

| Domain | Dataset | Classes | Train | Test | | Target Class |
|--------|---------|---------|-------|------|------|--------------|
| | | | | Clean | Poison | |
| Text | SST-2 | 2 | 67,349 | 872 | 444 | Negative (0) |
| | MNLI | 3 | 100,000 | 400 | 285 | Neutral (1) |
| | AGNews | 4 | 120,000 | 7,600 | 5,700 | Sports (1) |
| Vision | CIFAR-10 | 10 | 50,000 | 10,000 | 9,000 | Automobile (1) |
| | TinyImageNet | 200 | 100,000 | 10,000 | 9,950 | European Fire Salamander (1) |

### G.2  Dataset Licenses

We evaluate our method on the following datasets: **SST-2** [43], **MNLI** [54], **AG News** [66], **CIFAR-10** [21], and **TinyImageNet** [23].

The **MNLI** dataset is released under the Open American National Corpus (OANC) license, which permits free use, as stated in the original paper [54]. The **AG News** dataset is distributed with a disclaimer stating it is provided "as is" without warranties and does not impose explicit restrictions on academic use.[1] No public licensing information was found for **SST-2**, **CIFAR-10**, or **TinyImageNet**. We use all datasets solely for academic, non-commercial research purposes, in accordance with standard practice in the machine learning community.

### G.3  Defense Baselines

We evaluate seven defensive approaches across text and vision domains: three model-merging techniques common to both domains, plus two domain-specific data purification methods for each– one applied during training and another during inference.

The three model-merging methods are: (1) **TIES** [59], (2) **DARE** [63], and (3) **WAG** [2]. These methods are chosen because they are applicable to both text and vision domains, do not rely on assumptions about backdoor priors, and eliminate the need for large-scale proxy clean or compromised data used for model purification or retraining. Their alignment with our setting makes them suitable for comparison. For conventional baselines, we use **Z-Def.** [17] and **ONION** [37] in the text domain, which detect outlier trigger words during training and testing, respectively. For the vision domain, we select **CutMix** [64] and **ShrinkPad** [27]. CutMix mitigates backdoor attacks by mixing image patches, disrupting the spatial integrity of triggers. ShrinkPad defends by shrinking the image and padding it, altering trigger placement, and reducing its effectiveness. For the vision domain, we use the BackdoorBox toolkit [28] to apply these defenses. Specifically, for CutMix, we use 30 epochs to repair the model. While these well-established methods are representative in terms of usage and performance, their dependence on data access may limit practicality in some scenarios. All baseline methods use their open-source codebases with default hyperparameters.

### G.4  Experiment Resources

We conduct the model training and module switching experiments using three seeds on a single Nvidia A100 GPU , reporting the average performance. We run the evolutionary search for 2,000,000 generations on a CPU, which takes six hours for a given merging configuration (*e.g.,* two models with 24 layers and six modules per layer). This search only needs to be performed once, as the discovered strategy can serve as an artifact that applies to all future combinations of the same architecture.

---

[1] http://groups.di.unipi.it/~gulli/AG_corpus_of_news_articles.html

# H  Additional results

## H.1  Overall Defense Performance for Textual Backdoor Attacks

Due to space constraints, we present comprehensive experimental results for three datasets (**SST-2**, **MNLI**, and **AG News**) in Table 7, Table 8, and Table 9. All experiments follow the controlled settings described in Section 5.1, utilizing *RoBERTa-large* as the victim model, with results averaged across three random seeds.

We observe that our method yields decent performance on the **SST-2** dataset: it achieves top performance in 8 out of 10 attack combinations, with the remaining 2 combinations ranking second best. In cases where our method ranks first, it significantly outperforms baseline approaches. For instance, when combining BadNet with LWS attacks, our method achieves an average ASR score 21% lower than the second-best defense method. Moreover, our method consistently achieves the lowest individual ASR scores across both attacks in most combinations, highlighting its effectiveness in simultaneously mitigating multiple threats when merging compromised models.

Even in scenarios where our method ranks second, it maintains comparable defense performance to the top-performing approach. Furthermore, when combining clean models with compromised ones, our method demonstrates strong resistance against malicious attack injection, as evidenced by the lowest ASR scores. Notably, our method maintains good utility preservation across all combinations, showing minimal impact to the model performance.

Table 7: Performance comparison on the **SST-2** dataset using the *RoBERTa-large* model.

| Defense | CACC | BadNet | Insert | LWS | Hidden | AVG. | Defense | CACC | BadNet | Insert | LWS | Hidden | AVG. |
|---|---|---|---|---|---|---|---|---|---|---|---|---|---|
| Benign | 95.9 | 4.1 | 2.2 | 12.8 | 16.5 | 8.9 | Z-Def | 95.6* | 4.6 | 1.8 | 97.3 | 35.7 | 34.9 |
| Victim | 95.9* | 100.0 | 100.0 | 98.0 | 96.5 | 98.6 | ONION | 92.8* | 56.8 | 99.9 | 85.7 | 92.9 | 83.8 |
| *Combined: BadNet + InsertSent* | | | | | | | *Combined: InsertSent + LWS* | | | | | | |
| WAG | 96.3 | 56.3 | 7.4 | - | - | 31.9 | WAG | 96.1 | - | 15.1 | 43.3 | - | 29.2 |
| TIES | 95.9 | 88.7 | 17.0 | - | - | 52.9 | TIES | 96.1 | - | 35.8 | 64.9 | - | 50.3 |
| DARE | 96.5 | 57.8 | 36.3 | - | - | 47.1 | DARE | 96.4 | - | 44.4 | 31.5 | - | 37.9 |
| Ours | 96.2 | 36.9 | 7.1 | - | - | 22.0 | Ours | 96.0 | - | 11.9 | 39.7 | - | 25.8 |
| *Combined: BadNet + LWS* | | | | | | | *Combined: InsertSent + HiddenKiller* | | | | | | |
| WAG | 96.2 | 74.0 | - | 50.3 | - | 62.2 | WAG | 96.3 | - | 12.5 | - | 28.5 | 20.5 |
| TIES | 95.9 | 88.1 | - | 66.1 | - | 77.1 | TIES | 95.9 | - | 37.5 | - | 39.0 | 38.3 |
| DARE | 96.2 | 60.4 | - | 62.5 | - | 61.4 | DARE | 96.6 | - | 38.7 | - | 29.1 | 33.9 |
| Ours | 96.0 | 41.7 | - | 39.0 | - | 40.4 | Ours | 95.8 | - | 10.1 | - | 28.7 | 19.4 |
| *Combined: BadNet + HiddenKiller* | | | | | | | *Combined: LWS + HiddenKiller* | | | | | | |
| WAG | 96.1 | 63.9 | - | - | 29.0 | 46.4 | WAG | 96.4 | - | - | 60.5 | 41.7 | 51.1 |
| TIES | 96.0 | 90.4 | - | - | 36.9 | 63.6 | TIES | 96.0 | - | - | 77.8 | 55.8 | 66.8 |
| DARE | 96.7 | 36.3 | - | - | 47.6 | 41.9 | DARE | 96.7 | - | - | 67.7 | 43.3 | 55.5 |
| Ours | 96.1 | 40.5 | - | - | 27.7 | 34.1 | Ours | 96.0 | - | - | 58.6 | 47.2 | 52.9 |
| *Combined: Benign + BadNet* | | | | | | | *Combined: Benign + LWS* | | | | | | |
| WAG | 96.1 | 39.3 | - | - | - | 39.3 | WAG | 96.1 | - | - | 43.3 | - | 43.3 |
| TIES | 95.7 | 69.2 | - | - | - | 69.2 | TIES | 95.8 | - | - | 60.7 | - | 60.7 |
| DARE | 96.4 | 43.2 | - | - | - | 43.2 | DARE | 96.6 | - | - | 72.3 | - | 72.3 |
| Ours | 96.1 | 12.2 | - | - | - | 12.2 | Ours | 95.9 | - | - | 39.0 | - | 39.0 |
| *Combined: Benign + InsertSent* | | | | | | | *Combined: Benign + HiddenKiller* | | | | | | |
| WAG | 96.1 | - | 5.5 | - | - | 5.5 | WAG | 96.0 | - | - | - | 24.9 | 24.9 |
| TIES | 96.1 | - | 9.0 | - | - | 9.0 | TIES | 96.1 | - | - | - | 30.0 | 30.0 |
| DARE | 96.6 | - | 4.7 | - | - | 4.7 | DARE | 96.7 | - | - | - | 38.2 | 38.2 |
| Ours | 96.1 | - | 4.1 | - | - | 4.1 | Ours | 96.0 | - | - | - | 25.5 | 25.5 |

For the results of **MNLI** dataset Table 8, our method demonstrates more balanced and robust defense performance across different attack combinations. While DARE occasionally achieves lower ASR on individual attacks (*e.g.,* 11.6% ASR for BadNet in BadNet+InsertSent combination), it shows significant vulnerability to the other attack type (90.6% ASR for InsertSent), indicating potential risks when merging with new models. In contrast, our method maintains consistently lower average ASRs across various combinations (*e.g.,* 23.7% for BadNet+InsertSent, 43.7% for InsertSent+LWS, and 40.2% for InsertSent+Hidden), demonstrating its effectiveness in simultaneously defending against multiple attack types.

For the results of **AG NEWS** dataset Table 9, we observe a similar pattern, where our method provides more balanced defense capabilities. Notably, for the InsertSent+LWS combination, while DARE

achieves a low ASR of 1.2% on LWS, it remains highly vulnerable to InsertSent attacks (99.6% ASR). In contrast, our method maintains consistently lower ASRs for both attacks (9.5% and 16.7%), resulting in a better average performance of 13.1%.

Table 8: Performance comparison on the **MNLI** dataset using the *RoBERTa-large* model.

| Defense | CACC | BadNet | Insert | LWS | Hidden | AVG. | Defense | CACC | BadNet | Insert | LWS | Hidden | AVG. |
|---|---|---|---|---|---|---|---|---|---|---|---|---|---|
| Benign | 87.6 | 12.3 | 12.6 | 26.4 | 36.9 | 22.1 | Z-Def | 89.2* | 11.1 | 11.6 | 92.2 | 50.6 | 41.4 |
| Victim | 89.5* | 100.0 | 100.0 | 96.0 | 99.9 | 99.0 | ONION | 86.3* | 64.3 | 98.6 | 89.0 | 98.8 | 87.7 |
| *Combined: BadNet + InsertSent* | | | | | | | *Combined: InsertSent + LWS* | | | | | | |
| WAG | 90.3 | 39.8 | 27.6 | - | - | 33.7 | WAG | 90.6 | - | 36.1 | 62.6 | - | 49.4 |
| TIES | 90.3 | 73.6 | 56.1 | - | - | 64.9 | TIES | 90.3 | - | 60.0 | 65.3 | - | 62.7 |
| DARE | 91.3 | 11.6 | 90.6 | - | - | 51.1 | DARE | 91.4 | - | 88.8 | 40.2 | - | 64.5 |
| Ours | 90.5 | 24.8 | 22.5 | - | - | 23.7 | Ours | 91.0 | - | 24.8 | 62.5 | - | 43.7 |
| *Combined: BadNet + LWS* | | | | | | | *Combined: InsertSent + Hidden* | | | | | | |
| WAG | 89.8 | 59.3 | - | 69.3 | - | 64.3 | WAG | 91.5 | - | 36.6 | - | 46.9 | 41.8 |
| TIES | 90.0 | 87.3 | - | 73.1 | - | 80.2 | TIES | 90.9 | - | 65.1 | - | 55.2 | 60.2 |
| DARE | 90.5 | 71.7 | - | 56.4 | - | 64.1 | DARE | 91.8 | - | 90.8 | - | 40.2 | 65.5 |
| Ours | 90.1 | 45.1 | - | 68.9 | - | 57.0 | Ours | 91.1 | - | 24.3 | - | 56.1 | 40.2 |
| *Combined: BadNet + Hidden* | | | | | | | *Combined: LWS + Hidden* | | | | | | |
| WAG | 89.9 | 61.6 | - | - | 51.7 | 56.7 | WAG | 89.8 | - | - | 70.2 | 55.1 | 62.7 |
| TIES | 90.0 | 89.4 | - | - | 64.0 | 76.7 | TIES | 90.1 | - | - | 73.8 | 59.1 | 66.5 |
| DARE | 90.9 | 33.4 | - | - | 81.8 | 57.6 | DARE | 91.0 | - | - | 41.5 | 88.7 | 65.1 |
| Ours | 90.2 | 32.5 | - | - | 59.3 | 45.9 | Ours | 89.9 | - | - | 70.3 | 57.3 | 63.8 |
| *Combined: Benign + BadNet* | | | | | | | *Combined: LWS + Benign* | | | | | | |
| WAG | 90.2 | 47.8 | - | - | - | 47.8 | WAG | 89.0 | - | - | 65.6 | - | 65.6 |
| TIES | 89.8 | 64.9 | - | - | - | 64.9 | TIES | 89.8 | - | - | 69.3 | - | 69.3 |
| DARE | 91.0 | 41.8 | - | - | - | 41.8 | DARE | 90.1 | - | - | 48.9 | - | 48.9 |
| Ours | 90.1 | 43.3 | - | - | - | 43.3 | Ours | 89.3 | - | - | 64.1 | - | 64.1 |
| *Combined: InsertSent + Benign* | | | | | | | *Combined: Hidden + Benign* | | | | | | |
| WAG | 90.4 | - | 23.2 | - | - | 23.2 | WAG | 90.3 | - | - | - | 47.0 | 47.0 |
| TIES | 90.4 | - | 40.6 | - | - | 40.6 | TIES | 89.8 | - | - | - | 54.3 | 54.3 |
| DARE | 91.3 | - | 42.3 | - | - | 42.3 | DARE | 90.9 | - | - | - | 63.3 | 63.3 |
| Ours | 90.5 | - | 18.3 | - | - | 18.3 | Ours | 89.4 | - | - | - | 47.9 | 47.9 |

## H.2 Overall Defense Performance for Vision Backdoor Attacks

We present the full results for the **CIFAR-10** and **TinyImageNet** datasets with the ViT model in Table 10 and Table 11, respectively.

While most methods achieve relatively low ASRs for many attack types, our approach is particularly effective against stealthier attacks like PhysicalBA. This is most evident in the BadNet+PhysicalBA combination, where our method reduces the ASR to 18.5% for both attacks while maintaining a high clean accuracy of 98.7% in CIFAR-10 dataset. These results highlight our method's strength in defending against more sophisticated visual backdoor attacks.

## H.3 Fitness Score Comparison of Different Strategy

We investigate the defense performance using two different evolutionary search strategies, with and without early stopping, as illustrated in Figure 7 and 6, and present their fitness score breakdown in Table 12. The early stopping criterion terminates the search when no improvement in fitness score is observed over 100,000 iterations. We observe a positive correlation between the fitness score and defense performance: the adopted strategy without early stopping achieves a lower fitness score and reduces the ASR by 27.2%. By examining the score breakdowns and the visualized combinations, we attribute this improvement to fewer violations of residual connection rules in the adopted strategy, which helps disrupt subtle spurious correlations more effectively.

## H.4 Results of Candidate Selection

As our method asymmetrically allocates modules to models, a set of candidates is generated, for which we design a selection method illustrated in Section 4.4. While the chosen candidate consistently performs well, we analyze unselected candidates' performance, as shown in Table 13. Our selection method correctly identifies the best candidates in most cases, outperforming alternatives by

Table 9: Performance comparison on the **AG NEWS** dataset using the *RoBERTa-large* model.

| Defense | CACC | BadNet | Insert | LWS | Hidden | AVG. | Defense | CACC | BadNet | Insert | LWS | Hidden | AVG. |
|---|---|---|---|---|---|---|---|---|---|---|---|---|---|
| Benign | 95.4 | 1.9 | 0.5 | 0.5 | 1.1 | 1.0 | Z-Def | 95.4* | 1.6 | 0.4 | 97.9 | 100.0 | 50.0 |
| Victim | 95.0* | 99.9 | 99.6 | 99.6 | 100.0 | 99.8 | ONION | 92.3* | 59.4 | 97.8 | 84.8 | 99.6 | 85.4 |
| *Combined: BadNet + InsertSent* | | | | | | | *Combined: InsertSent + LWS* | | | | | | |
| WAG | 95.4 | 75.2 | 60.2 | - | - | 67.7 | WAG | 95.2 | - | 39.5 | 17.8 | - | 28.7 |
| TIES | 95.3 | 92.4 | 95.6 | - | - | 94.0 | TIES | 95.1 | - | 90.5 | 55.7 | - | 73.1 |
| DARE | 95.6 | 33.7 | 66.6 | - | - | 50.1 | DARE | 95.4 | - | 99.6 | 1.2 | - | 50.4 |
| Ours | 95.3 | 72.3 | 42.5 | - | - | 57.4 | Ours | 95.1 | - | 9.5 | 16.7 | - | 13.1 |
| *Combined: BadNet + LWS* | | | | | | | *Combined: InsertSent + Hidden* | | | | | | |
| WAG | 95.2 | 76.1 | - | 28.1 | - | 52.1 | WAG | 95.4 | - | 61.4 | - | 43.6 | 52.5 |
| TIES | 95.1 | 95.6 | - | 64.4 | - | 80.0 | TIES | 95.3 | - | 93.4 | - | 75.3 | 84.4 |
| DARE | 95.4 | 99.3 | - | 3.5 | - | 51.4 | DARE | 95.5 | - | 84.0 | - | 15.8 | 49.9 |
| Ours | 95.2 | 75.8 | - | 26.0 | - | 50.9 | Ours | 95.3 | - | 41.7 | - | 47.5 | 44.6 |
| *Combined: BadNet + Hidden* | | | | | | | *Combined: LWS + Hidden* | | | | | | |
| WAG | 95.2 | 73.2 | - | - | 37.2 | 55.2 | WAG | 95.1 | - | - | 31.7 | 62.6 | 47.2 |
| TIES | 95.3 | 91.9 | - | - | 71.9 | 81.9 | TIES | 95.1 | - | - | 67.5 | 92.2 | 79.9 |
| DARE | 95.4 | 66.7 | - | - | 40.4 | 53.6 | DARE | 95.3 | - | - | 2.5 | 99.9 | 51.2 |
| Ours | 95.2 | 56.5 | - | - | 38.1 | 47.3 | Ours | 95.2 | - | - | 33.5 | 60.5 | 47.0 |
| *Combined: Benign + BadNet* | | | | | | | *Combined: Benign + LWS* | | | | | | |
| WAG | 95.4 | 65.4 | - | - | - | 65.4 | WAG | 95.2 | - | - | 14.0 | - | 14.0 |
| TIES | 95.4 | 87.4 | - | - | - | 87.4 | TIES | 95.2 | - | - | 47.1 | - | 47.1 |
| DARE | 95.6 | 33.6 | - | - | - | 33.6 | DARE | 95.6 | - | - | 2.6 | - | 2.6 |
| Ours | 95.4 | 46.4 | - | - | - | 46.4 | Ours | 95.2 | - | - | 15.7 | - | 15.7 |
| *Combined: Benign + InsertSent* | | | | | | | *Combined: Benign + Hidden* | | | | | | |
| WAG | 95.4 | - | 56.6 | - | - | 56.6 | WAG | 95.3 | - | - | - | 36.4 | 36.4 |
| TIES | 95.3 | - | 93.2 | - | - | 93.2 | TIES | 95.3 | - | - | - | 68.8 | 68.8 |
| DARE | 95.6 | - | 3.1 | - | - | 3.1 | DARE | 95.5 | - | - | - | 7.4 | 7.4 |
| Ours | 95.3 | - | 16.6 | - | - | 16.6 | Ours | 95.3 | - | - | - | 48.0 | 48.0 |

a significant margin. Although some unselected candidates achieve a lower ASR in certain cases, our selected candidate maintains comparable performance.

## H.5 Importance of Heuristic Rules

We introduce five heuristic rules in Section 4.2 to guide the evolutionary search for module switching strategies. To assess the contribution of each rule, we perform ablation experiments by individually removing the first three rules, which aim to disconnect adjacent modules at different structural levels, and measure the resulting defense performance under three settings. As shown in Table 14, removing any of these rules generally leads to performance degradation, supporting the complementary nature of the full rule set. We further visualize the searched strategies resulting from each ablation in Figures 8 to 10.

## H.6 Generalization across Model Architectures

We evaluate our method across three model architectures–*RoBERTa-large*, *BERT-large*, and *DeBERTa-v3-large*–under three backdoor settings. As shown in Table 15, our defense consistently achieves lower ASR compared to the baseline WAG across all models. Notably, we apply the same unified searched strategy (presented in Figure 6) to all architectures, demonstrating the strong generalization and transferability of our method. This supports its scalability and practicality in real-world applications.

## H.7 Minimum Clean Data Requirement

By default, we use 50 clean data points per class to guide the candidate selection process (as described in Section 4.4). To further investigate the minimum clean data required for effective defense, we reduce this to 20 samples per class across all three model architectures on SST-2. As shown in Table 15, our approach continues to select candidates with low ASR even under this constrained setting. These results indicate that the method remains effective in low-resource scenarios with limited clean supervision.

Table 10: Performance comparison on the **CIFAR-10** dataset using the *ViT* model.

| Defense | CACC | BadNet | WaNet | BATT | PBA | AVG. | Defense | CACC | BadNet | WaNet | BATT | PBA | AVG. |
|---|---|---|---|---|---|---|---|---|---|---|---|---|---|
| Benign | 98.8 | 10.1 | 10.2 | 7.7 | 10.1 | 9.5 | CutMix | 97.7* | 87.1 | 70.6 | 99.9 | 64.9 | 80.6 |
| Victim | 98.5* | 96.3 | 84.7 | 99.9 | 89.4 | 92.6 | ShrinkPad | 97.3* | 14.4 | 51.3 | 99.9 | 88.3 | 63.5 |
| *Combined: BadNet + WaNet* | | | | | | | *Combined: WaNet + BATT* | | | | | | |
| WAG | 98.7 | 13.8 | 10.6 | - | - | 12.2 | WAG | 98.7 | - | 10.2 | 22.3 | - | 16.3 |
| TIES | 98.6 | 11.9 | 10.6 | - | - | 11.3 | TIES | 98.9 | - | 10.2 | 23.9 | - | 17.0 |
| DARE | 98.8 | 83.3 | 10.2 | - | - | 46.7 | DARE | 98.9 | - | 10.2 | 45.8 | - | 28.0 |
| Ours | 98.7 | 12.3 | 10.5 | - | - | 11.4 | Ours | 98.7 | - | 10.3 | 19.1 | - | 14.7 |
| *Combined: BadNet + BATT* | | | | | | | *Combined: WaNet + PhysicalBA* | | | | | | |
| WAG | 98.9 | 10.1 | - | 42.7 | - | 26.4 | WAG | 98.8 | - | 10.2 | - | 10.2 | 10.2 |
| TIES | 98.9 | 10.1 | - | 55.8 | - | 33.0 | TIES | 98.9 | - | 10.1 | - | 10.3 | 10.2 |
| DARE | 99.0 | 69.2 | - | 26.8 | - | 48.0 | DARE | 98.9 | - | 10.1 | - | 21.0 | 15.6 |
| Ours | 98.7 | 10.2 | - | 32.6 | - | 21.4 | Ours | 98.7 | - | 10.3 | - | 10.2 | 10.2 |
| *Combined: BadNet + PhysicalBA* | | | | | | | *Combined: BATT + PhysicalBA* | | | | | | |
| WAG | 99.0 | 39.5 | - | - | 39.5 | 39.5 | WAG | 98.9 | - | - | 26.8 | 10.0 | 18.4 |
| TIES | 98.9 | 43.1 | - | - | 43.1 | 43.1 | TIES | 98.7 | - | - | 23.4 | 10.0 | 16.7 |
| DARE | 99.0 | 72.2 | - | - | 72.2 | 72.2 | DARE | 98.9 | - | - | 23.0 | 10.1 | 16.5 |
| Ours | 98.7 | 18.5 | - | - | 18.4 | 18.5 | Ours | 98.8 | - | - | 9.8 | 10.0 | 9.9 |
| *Combined: Benign + BadNet* | | | | | | | *Combined: Benign + WaNet* | | | | | | |
| WAG | 98.8 | 19.4 | - | - | - | 19.4 | WAG | 98.9 | - | 10.2 | - | - | 10.2 |
| TIES | 98.8 | 10.2 | - | - | - | 10.2 | TIES | 98.6 | - | 10.3 | - | - | 10.3 |
| DARE | 98.8 | 10.3 | - | - | - | 10.3 | DARE | 98.8 | - | 10.2 | - | - | 10.2 |
| Ours | 98.7 | 10.3 | - | - | - | 10.3 | Ours | 98.7 | - | 10.3 | - | - | 10.3 |
| *Combined: Benign + BATT* | | | | | | | *Combined: Benign + PhysicalBA* | | | | | | |
| WAG | 98.8 | - | - | 19.4 | - | 19.4 | WAG | 99.0 | - | - | - | 10.1 | 10.1 |
| TIES | 98.8 | - | - | 23.4 | - | 23.4 | TIES | 98.8 | - | - | - | 10.2 | 10.2 |
| DARE | 99.0 | - | - | 28.2 | - | 28.2 | DARE | 99.9 | - | - | - | 10.1 | 10.1 |
| Ours | 98.8 | - | - | 15.8 | - | 15.8 | Ours | 98.9 | - | - | - | 10.1 | 10.1 |

Table 11: Performance comparison on the **TinyImageNet** dataset using the *ViT* model.

| Defense | CACC | BadNet | WaNet | BATT | PBA | AVG. |
|---|---|---|---|---|---|---|
| Benign | 89.1 | 0.51 | 0.01 | 0.04 | 0.03 | 0.15 |
| Victim | 85.8* | 97.8 | 98.9 | 100.0 | 90.0 | 96.6 |
| *Combined: BadNet + WaNet* | | | | | | |
| WAG | 88.2 | 11.7 | 5.5 | - | - | 8.6 |
| Ours | 84.2 | 0.6 | 0.2 | - | - | 0.4 |
| *Combined: BadNet + BATT* | | | | | | |
| WAG | 87.3 | 0.11 | - | 0.15 | - | 0.13 |
| Ours | 86.8 | 0.03 | - | 0.07 | - | 0.05 |
| *Combined: BadNet + PhysicalBA* | | | | | | |
| WAG | 88.5 | 58.5 | - | - | 35.9 | 47.2 |
| Ours | 84.8 | 48.2 | - | - | 29.1 | 38.7 |

## H.8 Performance under Varying Poisoning Rates

We further evaluate the robustness of our method under varying poisoning rates (20%, 10%, and 1%) on SST-2 dataset using the *RoBERTa-large* model. As shown in Table 16, our method consistently achieves lower ASR than WAG across settings that combine models poisoned with different attack methods and poisoning ratios.

Table 12: Comparison of strategy fitness scores and performance in combining *Benign* with *BadNet* model.

| Early Stopping Strategy | | Adopted Strategy | |
|---|---|---|---|
| *Fitness Score Components* | | | |
| Intra Layer Score | **-42.00** | Intra Layer Score | -48.00 |
| Inter Layer Score | -21.00 | Inter Layer Score | **-15.00** |
| Residual Connection Score | -48.24 | Residual Connection Score | **-24.02** |
| Balance Score | 0.00 | Balance Score | 0.00 |
| Diversity Score | **17.00** | Diversity Score | 12.00 |
| **Total Score** | -94.24 | **Total Score** | **-75.01** |
| *Performance Metrics* | | | |
| CACC (↑) | 96.70 | CACC (↑) | 96.10 |
| ASR (↓) | 39.40 | ASR (↓) | **12.20** |

Table 13: Performance comparison of selected and unselected candidates on **SST-2**.

| Setting | Selection candidate | | Unselected candidate | | Overall Mean ASR (↓) | WAG Mean ASR (↓) |
|---|---|---|---|---|---|---|
| | CACC (↑) | AVG. ASR (↓) | CACC (↑) | AVG. ASR (↓) | | |
| BadNet+InsertSent | 96.2 | **22.0** | 96.5 | 31.2 | 26.6 | 31.9 |
| BadNet+LWS | 96.0 | **40.4** | 95.9 | 72.4 | 56.4 | 62.2 |
| BadNet+Hidden | 96.1 | **34.1** | 96.0 | 48.5 | 41.3 | 46.5 |
| InsertSent+LWS | 96.0 | **25.8** | 96.0 | 30.3 | 28.1 | 29.2 |
| InsertSent+Hidden | 95.8 | 19.4 | 96.1 | **19.2** | 19.3 | 20.5 |
| LWS+Hidden | 96.0 | 52.9 | 96.2 | **49.6** | 51.3 | 51.1 |
| Average | 96.0 | **32.4** | 96.1 | 41.9 | 37.2 | 40.2 |

Table 14: Impact of heuristic rule ablations under different combinations of backdoor settings on **SST-2** using the *RoBERTa-large* model. Δ denotes the change in average ASR relative to the full rule set.

| Setting | Ablation | CACC (↑) | ASR (↓) | | | |
|---|---|---|---|---|---|---|
| | | | Atk1 | Atk2 | AVG. | Δ |
| BadNet + InsertSent | All rules (full) | 96.2 | 36.9 | 7.1 | **22.0** | – |
| | w/o rule 1 | 96.0 | **33.2** | 18.7 | 25.9 | +3.9 |
| | w/o rule 2 | 96.3 | 60.6 | 14.1 | 37.3 | +15.3 |
| | w/o rule 3 | 96.3 | 43.1 | **6.2** | 24.6 | +2.6 |
| BadNet + LWS | All rules (full) | 96.0 | **41.7** | **39.0** | **40.4** | – |
| | w/o rule 1 | 95.9 | 46.2 | 51.2 | 48.7 | +8.3 |
| | w/o rule 2 | 96.0 | 68.1 | 62.8 | 65.4 | +25.0 |
| | w/o rule 3 | 96.0 | 69.1 | 46.3 | 57.7 | +17.3 |
| BadNet + Hidden | All rules (full) | 96.1 | 40.5 | **27.7** | 34.1 | – |
| | w/o rule 1 | 95.9 | **14.0** | 32.8 | **23.4** | -10.7 |
| | w/o rule 2 | 96.1 | 59.4 | 29.4 | 44.4 | +10.3 |
| | w/o rule 3 | 96.0 | 56.6 | 29.1 | 42.9 | +8.8 |

Table 15: Cross-model evaluation under varying clean data budgets on **SST-2**. $N = 50$ and $N = 20$ indicate the number of clean samples per class used for validation.

| Setting | Defense | RoBERTa-large CACC (↑) | ASR (↓) Atk1 | Atk2 | AVG. | BERT-large CACC (↑) | ASR (↓) Atk1 | Atk2 | AVG. | DeBERTa-v3-large CACC (↑) | ASR (↓) Atk1 | Atk2 | AVG. |
|---|---|---|---|---|---|---|---|---|---|---|---|---|---|
| BadNet + InsertSent | WAG | 96.3 | 56.3 | 7.4 | 31.9 | 93.3 | 40.2 | 60.1 | 50.2 | 96.1 | 47.4 | 5.2 | 26.3 |
| | Ours ($N = 50$) | 96.2 | **36.9** | 7.1 | **22.0** | 93.5 | 39.7 | 38.1 | 38.9 | 96.3 | 40.4 | 5.2 | 22.8 |
| | Ours ($N = 20$) | 96.2 | 47.7 | **6.6** | 27.1 | 93.5 | **39.7** | **38.1** | **38.9** | 96.3 | **32.8** | **5.1** | **19.0** |
| BadNet + LWS | WAG | 96.2 | 74.0 | 50.3 | 62.2 | 93.1 | 76.9 | 63.0 | 69.9 | 96.2 | 63.4 | 79.5 | 71.5 |
| | Ours ($N = 50$) | 96.0 | **41.7** | **39.0** | **40.4** | 93.0 | **73.9** | **61.3** | **67.6** | 96.0 | **48.7** | **73.0** | **60.8** |
| | Ours ($N = 20$) | 96.0 | **41.7** | **39.0** | **40.4** | 93.0 | 76.5 | 63.6 | 70.0 | 96.0 | **48.7** | **73.0** | **60.8** |
| BadNet + Hidden | WAG | 96.1 | 63.9 | 29.0 | 46.5 | 93.3 | 56.9 | 43.8 | 50.3 | 96.2 | 48.3 | **39.6** | 43.9 |
| | Ours ($N = 50$) | 96.1 | **40.5** | 27.7 | **34.1** | 93.4 | **50.3** | **37.9** | **44.1** | 96.1 | **22.7** | 41.0 | **31.8** |
| | Ours ($N = 20$) | 96.2 | 34.9 | **25.6** | 30.3 | 93.4 | **50.3** | **37.9** | **44.1** | 96.3 | **22.7** | 41.0 | **31.8** |

Table 16: Performance comparison under varying poison rates on **SST-2** using the *RoBERTa-large* model.

| Setting | Defense | Poison Rate: 20% CACC (↑) | ASR (↓) Atk1 | Atk2 | AVG. | Poison Rate: 10% CACC (↑) | ASR (↓) Atk1 | Atk2 | AVG. | Poison Rate: 1% CACC (↑) | ASR (↓) Atk1 | Atk2 | AVG. |
|---|---|---|---|---|---|---|---|---|---|---|---|---|---|
| BadNet + InsertSent | WAG | 96.3 | 56.3 | 7.4 | 31.9 | 96.1 | 66.6 | **8.9** | 37.9 | 96.4 | 58.3 | **27.2** | **42.8** |
| | Ours (MSD) | 96.2 | **36.9** | **7.1** | **22.0** | 96.0 | **55.1** | 9.3 | **32.3** | 96.3 | **57.4** | 44.4 | 50.9 |
| BadNet + LWS | WAG | 96.2 | 74.0 | 50.3 | 62.2 | 95.1 | 83.7 | 46.3 | 65.0 | 96.3 | 62.7 | 28.9 | 45.8 |
| | Ours (MSD) | 96.0 | **41.7** | **39.0** | **40.4** | 94.9 | **70.6** | **40.1** | **55.3** | 96.4 | **59.9** | **27.6** | **43.7** |
| BadNet + Hidden | WAG | 96.1 | 63.9 | 29.0 | 46.5 | 95.9 | 67.9 | 26.9 | 47.4 | 96.1 | 64.9 | 30.5 | 47.7 |
| | Ours (MSD) | 96.1 | **40.5** | **27.7** | **34.1** | 95.5 | **51.9** | **25.8** | **38.9** | 96.1 | **59.2** | **30.0** | **44.6** |

# I Examples of Searched Strategy

We present several examples of module switching strategies discovered by our evolutionary algorithm, listed as follows:

- Our adopted merging strategy for two-model combinations using *RoBERTa-large* (24 layers), presented in Figure 6, achieves a fitness score of -75.0.
- An early-stage merging strategy for *RoBERTa-large* (24 layers), shown in Figure 7, yields a fitness score of -94.2.
- The adopted strategy for merging three *RoBERTa-large* models (24 layers), illustrated in Figure 11, obtains a fitness score of -26.2.
- An alternative merging strategy designed for *ViT* model (12 layers), depicted in Figure 12, achieves a fitness score of -39.5.

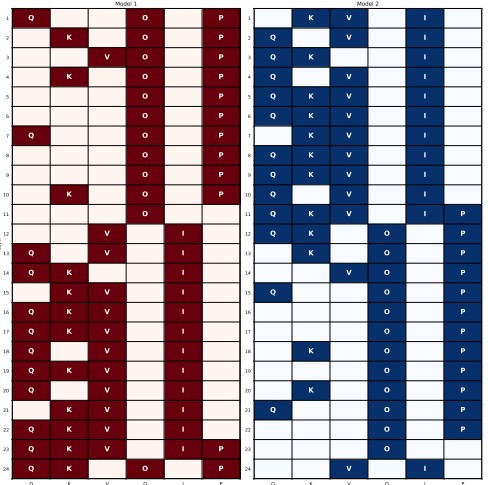

Figure 6: Adopted merging strategy (with a fitness score of -75.0).

Figure 7: Early stopping strategy (with a fitness score of -94.2).

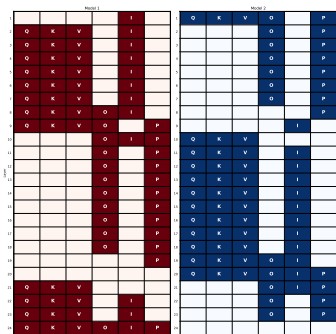
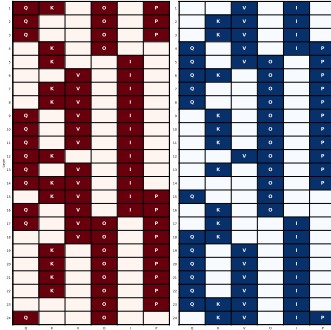
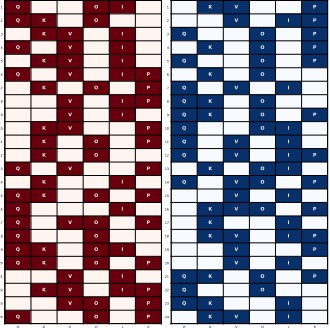

Figure 8: Strategy of ablating rule 1.

Figure 9: Strategy of ablating rule 2.

Figure 10: Strategy of ablating rule 3.

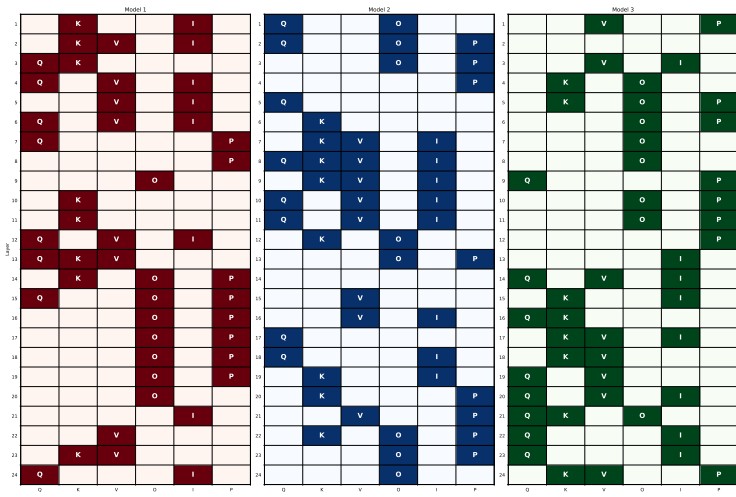

Figure 11: Adopted merging strategy (fitness score -26.2).

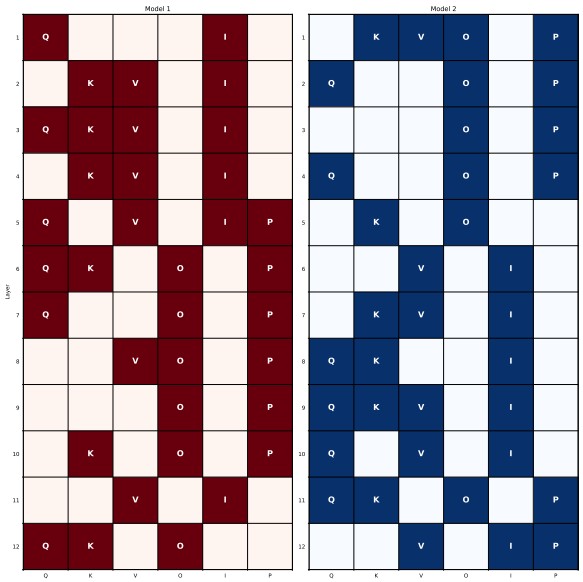

Figure 12: Adopted merging strategy (fitness score -39.5).

