# OpenReview forum: "Defending Deep Neural Networks against Backdoor Attacks via Module Switching"
_NeurIPS.cc/2025/Conference — Submitted to NeurIPS 2025_

### Official Review · Reviewer_uUUc · 2025-06-29

**Clarity:** 3
**Significance:** 2
**Originality:** 3
**Rating:** 4
**Confidence:** 4

**Summary:**

This paper proposes a model-switching strategy for defending against backdoor attacks. It focuses on transformer-based models. It leverages an evolutionary algorithm to find a switching strategy to select different components from different models that can potentially break the learned shortcut in one backdoored model. Experiments on three NLP datasets and two vision datasets show that it can maintain high benign utility while reducing the backdoor attack success rate.

**Questions:**

Please refer to the weaknesses.

**Ethical Concerns:**

["NO or VERY MINOR ethics concerns only"]

**Final Justification:**

Thank the authors for the rebuttal. I increased my score.

**Limitations:**

The overhead is high. It's unclear about the effectiveness on other architectures.

**Paper Formatting Concerns:**

I didn't notice any.

**Quality:**

2

**Strengths And Weaknesses:**

Strengths:
1. It studies an important problem in defense against backdoor attacks.
2. The idea to merge backdoored models to break the learned backdoor shortcut is interesting.
3. The experimental results show that it outperforms several baselines.

Weaknesses:
1. The theoretical analysis is based on a two-layer network without non-linearity. It's suggested to add some theoretical analysis involving the non-linearity.
2. Additionally, the analysis and experiments assume that the backdoor models are fine-tuned from the same pre-trained model. This is a very strong assumption that may not be satisfied in the real world.
3. Similarly, the backdoored models in the experiments all have the same target class, and with different backdoor attacks. This may not hold in the real world. It would be beneficial to evaluate merging 1) backdoor models that are attacked by the same backdoor attacks and the same or different target classes, and 2) backdoor models that are attacked with different backdoor attacks and different target classes.
4. The analysis doesn't consider the benign utility (i.e., CACC) and thus cannot provide any guarantee on it. The model-switching algorithm doesn't consider the benign utility either. It's unclear to me why the benign utility wasn't affected as shown by the experiments.
5. The overhead is very high. It takes 6 hours to find a switching strategy.
6. It's unclear whether the model-switch method can be applied to other architectures beyond transformers.
7. It lacks the evaluation of other attacks and defenses, such as [Combat](https://ojs.aaai.org/index.php/AAAI/article/view/28019), [NARCISSUS](https://arxiv.org/pdf/2204.05255),  [FST](https://proceedings.neurips.cc/paper_files/paper/2023/file/ee37d51b3c003d89acba2363dde256af-Paper-Conference.pdf), [UNICORN](/forum?id=Mj7K4lglGyj), [MEDIC](https://openaccess.thecvf.com/content/CVPR2023/papers/Xu_MEDIC_Remove_Model_Backdoors_via_Importance_Driven_Cloning_CVPR_2023_paper.pdf), [NC](https://ieeexplore.ieee.org/document/8835365), [TanH](https://openaccess.thecvf.com/content/CVPR2022/papers/Tao_Better_Trigger_Inversion_Optimization_in_Backdoor_Scanning_CVPR_2022_paper.pdf) and [UNIT](https://arxiv.org/abs/2407.11372),
8. The empirical study at Line 137 doesn't include any training and leverages random noise. It would be better to really train those models.
9. Why is one switching strategy sufficient for different attacks? Also, it's suggested to conduct the adaptive attack where the attacker knows the switching strategy.
10. The NLP tasks are simple with 2-4 classes.

---

> ### Author Rebuttal · Authors · 2025-07-31
>
> We thank the reviewer for their insightful feedback. Below, we address each point in detail.
>
> ---
>
> **W1:** The theoretical analysis is based on a two-layer network without non-linearity; suggest including non-linearity analysis.
>
> **A1:** Our theoretical analysis focuses on the linear activation scenario, and empirical results show that non-linear activations in Figure 5 exhibit the same trend: switched models consistently show greater backdoor deviation than WAG models.
>
> We hypothesize this occurs because parameter updates ($\Delta$) are much smaller than original weights ($W$) in backdoor fine-tuning. For ReLU: $f(x) = W_2 \cdot \text{ReLU}(W_1 \cdot x) = W_2 \cdot (M \odot (W_1 \cdot x))$, where $M$ is a binary mask. When $|\Delta| \ll |W|$, the sign pattern remains largely unchanged, making local behavior effectively linear under a shared mask.
>
> Empirically, across four backdoored RoBERTa-large tasks, we observe $|W|$ = 65.8 $\pm$ 27.2, $|\Delta|$ = 1.28 $\pm$ 0.40, and $|\Delta|/|W|$ = 2.11% $\pm$ 0.46% for FFN weights. This small update magnitude supports the applicability of our linear analysis to non-linear networks and explains the consistent empirical trends.
>
> ---
>
> **W2:** The analysis assumes all backdoor models are fine-tuned from the same pretrained model — a strong assumption that may not hold in practice.
>
> **A2:** We clarify that fine-tuning from a shared pre-trained model is a standard and widely adopted practice. This is supported by Hugging Face stats: over 4,500 models are fine-tuned from `roberta-large`, and 3,600+ from `vit-base`. The `roberta-large` alone was downloaded over 1.2 million times monthly.
>
> Given the prevalence of this fine-tuning practice and the additional security benefits offered by model merging, we believe this setting is realistic and likely to become even more popular and widely adopted in future.
>
> ---
>
> **W3:** Experiments use backdoored models with the same target class but different attacks. Suggest evaluating: (1) same attack with same/different target classes, and (2) different attacks with different target classes.
>
> **A3:** We test both cases and find that our method works much better than WAG when the involved models have different target classes. Due to space constraints, the detailed results are provided in our responses A1 and A2 to reviewer `utYj`.
>
> These results confirm that our method remains robust even when the backdoored models differ in both attack types and target labels. In the special case where both models are attacked using the same method with the same target label, the shared bias cannot be removed, but this is a limitation that also applies to WAG, as averaging or switching has no effect when both models encode the same bias.
>
> We believe the increasing diversity of real-world datasets may help reduce such cases by lowering the likelihood of models sharing identical backdoor biases.
>
> ---
>
> **W4:** The analysis and model-switching algorithm do not consider benign utility. Unclear why benign utility remains unaffected in experiments.
>
> **A4:** We clarify benign utility preservation from three aspects: (1) theoretical and empirical analysis in our manuscript, (2) prior literature, and (3) additional experimental results on switching domain-relevant models.
>
> First, as shown in Section 3, since $M_i$ and $M_j$ share the same pretraining, their semantic knowledge is preserved by both weight averaging and our switching method, thereby maintaining benign utility. This is evidenced in Figures 2 and 5, where the WAG and Switched models remain close to the benign model in semantic space. As noted in W1, fine-tuning introduces only minor changes to pretrained weights.
>
> Second, prior studies [2, 60, 63] show that merging models—either through averaging or selective strategies—can preserve task performance. DARE [63] further shows that even after dropping parameters from cross-task but homologous models, the merged model performs well on both tasks (Figure 2).
>
> Third, we empirically validate this by merging models trained on stylistically different datasets (IMDB and SST-2). Our method maintains high benign accuracy (CACC) while greatly lowering ASR:
>
> |No.|Setting|SST-2 CACC|SST-2 Atk1 ASR|SST-2 Atk2 ASR|AVG. ASR|
> |-|-|-|-|-|-|
> |1|IMDB-Badnet|88.6|91.8|||
> |2|SST2-Sent|96.3||100.0||
> |3|WAG|95.5|18.3|57.2|37.8|
> |4|Ours|95.5|19.5|33.3|26.4|
>
> These results confirm that our method not only reduces backdoor vulnerabilities but also preserves benign task performance.
>
> ---
>
> **W5:** The overhead is very high. It takes 6 hours to find a switching strategy.
>
> **A5:** We clarify that the search needs to be performed only once per architecture, as the resulting strategy is transferable across models, modalities, and tasks. End users can directly apply the released strategy without incurring this cost.
>
> Importantly, the search is conducted entirely on CPUs and does not require high-performance hardware such as GPUs. We believe this makes the method accessible and practical. We acknowledge exploring more efficient search strategies as valuable future work.
>
> ---
>
> **W6:** It's unclear whether the model-switch method can be applied to other architectures beyond transformers.
>
> **A6:** We use Transformers as our testbed due to their relevance to our primary baselines and strong performance across both text and vision tasks [A]. While we have not evaluated CNNs or emerging architectures, we acknowledge this limitation in our submission and consider it a valuable direction for future work.
>
> As discussed in Section 4.2, we hypothesize that backdoors propagate through both feed-forward and residual connection paths—common components of modern neural networks (NNs). Consider a general NN structure:
>
> $y = F_{W_n}(\cdots F_{W_2}(F_{W_1}(x_0) + x_0) + x_1 \cdots ) + x_{n-1}, \quad x_i = F_{W_i}(x_{i-1}) + x_{i-1}$
>
> The key idea is to prevent adjacent weight modules from originating from the same model. In CNNs or MLPs, this could involve layer-level disruption or finer-grained kernel-level decomposition—directions for future exploration.
>
> ---
>
> **W7:** Lacks evaluation of other attacks and defenses, such as Combat, UNIT, and others.
>
> **A7:** Thank you for this suggestion. Our experimental design, which includes nine distinct experiments, focuses on validating our method's performance on Transformer architectures across both text and vision domains, building directly on the WAG baseline.
>
> Regarding the recommended methods, while most focus solely on CNNs, we did investigate the two relevant Transformer-based methods:
>
> * **UNIT:** Operates under a different assumption, requiring a large clean dataset (e.g., 5% of training data, i.e., 2,500 samples for CIFAR-10), which contrasts with our low-data setting of 20–50 samples per class.
> * **Combat:** Our evaluation of the released code for ViT-Base on CIFAR-10 resulted in low clean accuracy (70%), a trend consistent with the original paper (Table 3). Further adaptation would be needed for a fair comparison.
>
> We believe our current experiments provide a focused and comprehensive validation. We will add more detailed related work discussion in the revision.
>
> ---
>
> **W8:** The empirical study at Line 137 doesn't include any training and leverages random noise.
>
> **A8:** We clarify that our study involves proper training in two phases:
>
> * **Pretraining:** The network is trained on 250K synthetic samples $(X, S \times X)$ using MSE loss, where $X$ is random input and $S$ the weight matrix.
> * **Backdoor fine-tuning:** The model is then fine-tuned on 1,000 backdoor samples with outputs $(S + B) \times X$, where $B$ encodes the backdoor pattern.
>
> We repeat this process 1,000 times to generate distinct networks, enabling robust analysis. Also, we test four activation functions and observe consistent results, supporting its generality.
>
> ---
>
> **W9:** Why is one switching strategy sufficient for different attacks? Consider adaptive attacks where attacker knows strategy.
>
> **A9:** Our method does not rely on identifying attack-specific shortcut patterns. Instead, it recombines modules from different sources to cover as many shortcut paths as possible, disrupting backdoor behavior across diverse attacks.
>
> As a result, multiple switching strategies are effective—our method provides a systematic way to discover them. Figures 6 and 7 show two such strategies (one strong, one suboptimal), and additional runs with different seeds yield similarly effective variants.
>
> This diversity offers robustness against adaptive attacks. Even if the attacker knows one strategy (e.g., by freezing unused modules), the defender can apply another to break the transmission path.
>
> To validate this, we simulate an adaptive attacker aware of Strategy 1 (S1) and defend using Strategies 2 (S2) and 3 (S3). Results are shown below:
>
> |No.|Setting|CACC|Atk1 ASR|Atk2 ASR|AVG. ASR|
> |-|-|-|-|-|-|
> |1|Badnet|96.0|100.0|-|-|
> |2|Sent|96.3|-|100.0|-|
> |3|Adaptive Badnet (knows S1)|95.6|100.0|-|-|
> |4|Merge 1&2 (WAG)|96.7|56.3|7.4|31.9|
> |5|Ours–S2|96.0|**39.0**|8.0|**23.5**|
> |6|Ours–S3|96.0|**39.4**|20.6|**30.0**|
> |7|Merge 2&3 (WAG)|95.8|59.8|9.2|34.5|
> |8|Ours–S2|96.1|**21.3**|**7.2**|**14.3**|
> |9|Ours–S3|96.0|**57.5**|**5.4**|**31.5**|
>
> These results show that strategy diversity enables our method to remain effective under adaptive attacks.
>
> ---
>
> **W10:** The NLP tasks are simple with 2-4 classes.
>
> **A10:** We clarify that having 2–4 classes is typical for many NLP classification tasks and is a standard setting in the backdoor attack and defense literature. In fact, all 5 of the most recent works on text backdoor defenses ([2], [3], [68], [A], [B], cited in A2 to reviewer DvkL) adopt similar class settings. We follow these setups to enable fair comparisons.
>
> Moreover, the datasets used in our study—SST-2, MNLI, and AG News—are widely recognized benchmarks with strong community adoption, with high citation counts (10K, 5K, and 8K). Their use ensures relevance and rigor in evaluating backdoor defenses.

---

> > ### Comment · Reviewer_uUUc · 2025-08-04
> >
> > I appreciate the authors' effort in the detailed response. However, I still have some concerns.
> >
> > W2. I am still not convinced by the assumption that backdoored models are fine-tuned from the same pretrained model. The example provided by the authors is a single language model, which cannot be generalized to other models.
> >
> > W4. From the reply, the preservation of the benign utility is based on the assumption that the backdoored models are trained from the same model, and they preserve the benign utility in the same way, so that the defense doesn't explicitly constrain the benign utility. On the one hand, similar to W2, this is a strong assumption. On the other hand, different backdoor methods may not preserve the benign semantics in the same way, even if the underlying model is the same. For example, the backdoor attack can shuffle some layers so that after switching or averaging will fail to maintain the benign utility.
> >
> > W5. Why can't this search be executed on GPUs?
> >
> > W6. It appears that the method is not applicable to other networks besides transformers. As the title targets deep neural networks instead of transformers, it may be overstated.

---

> ### Author Response · Authors · 2025-08-04
>
> Thank you for these insightful follow-up comments. We appreciate your continued engagement and the opportunity to further clarify our assumptions and address the remaining concerns.
>
> ---
>
> **A2.** While in our initial rebuttal we listed `roberta-large` and `vit-base` as representative examples to show that practitioners often fine-tune models from the same pretrained backbone, we would like to clarify that **this practice generalizes to many other models**. For instance, the Hugging Face Hub includes over 14,000 models fine-tuned from `bert-base`, over 14,000 from `distilbert-base`, and over 3,600 from `deberta-v3`, with similar trends observed for many other foundation models. More importantly, as a proactive defender, **the user has the flexibility to choose the defense method and to select which models to use**, making this assumption both realistic and actionable.
>
> ---
>
> **A4.** Regarding the reviewer’s suggested case of adversarial manipulations such as layer shuffling: while it is possible in practice, to our knowledge, such operations are rarely discussed or evaluated in prior work.
>
> To further explore this non-trivial setting, we tested random layer shuffling on `roberta-large`, shuffling 2/4/6/12/24 layers. The clean accuracy dropped from 96.3% to 56.4%, 59.7%, 52.4%, 49.1%, and 47.2%, which are close to random guess performance. **These results show that such manipulations severely degrade clean performance**—contrary to the goal of stealthy backdoor attacks, which aim to preserve utility and avoid detection.
>
> We welcome the reviewer’s insights on how adversarial attackers might design more adaptive strategies that maintain clean accuracy while diverging from standard training behavior, and we are happy to further investigate such cases.
>
> ---
>
> **A5.** Our method can run on both CPU and GPU, but there is no significant difference in runtime for strategy search, as it does not involve loading models or performing heavy computations such as matrix multiplications. Considering the cost and accessibility, CPU is the more preferable setting. Prior search-based model combination methods (e.g., DARE [63] and DAM [60]) typically require two steps: (1) loading models and (2) verifying their performance on validation sets—**resulting in high computational overhead that must be incurred for every execution**. In contrast, our method is independent of specific model parameters and only needs to be executed once per architecture, making it more practical for scalable deployment.
>
> ---
>
> **A6.** Thank you for the comment. We would like to clarify that our work has explored various architectures, ranging from synthetic two-layer networks with varying activation functions to both text and vision Transformers. These represent both simple and advanced neural architectures, showing the generalization of our method and offering an interpretable perspective.

---

### Official Review · Reviewer_Jnww · 2025-07-01

**Clarity:** 3
**Significance:** 3
**Originality:** 4
**Rating:** 4
**Confidence:** 4

**Summary:**

This paper first introduces the perspective that module swapping outperforms weight averaging and provides a theoretical justification under the two-layer neural network setting. Building on this insight, the authors design a novel module-switching strategy that employs an evolutionary algorithm to eliminate the “shortcuts” exploited by backdoor attacks. Extensive experimental results demonstrate the effectiveness of the proposed method.

**Questions:**

Please see Strengths and Weaknesses. I am willing to raise the score if my questions are properly addressed.

**Ethical Concerns:**

["NO or VERY MINOR ethics concerns only"]

**Final Justification:**

The applicability of the method is somewhat limited. Therefore, I will maintain my score.

**Limitations:**

Yes.

**Paper Formatting Concerns:**

None.

**Quality:**

3

**Strengths And Weaknesses:**

Strengths

1. Through theoretical analysis, the superiority of the module switching over weight aggregation is demonstrated.
2. To identify the optimal switching strategy, a heuristic scoring function and an evolutionary algorithm are designed.
3. Extensive experiments are conducted to validate the effectiveness of the proposed method.

Weaknesses

1. The authors introduce an assumption that backdoor training is a two-stage process, which corresponds to the use of pre-trained models in the experiments. However, this assumption does not always hold in real-world scenarios. Backdoor attacks can be launched at the very beginning of training, in which case the model does not go through a benign phase. The authors should clarify how their method performs under such circumstances and provide experimental results under this setting.
2. The presentation of Figure 3 is somewhat confusing and needs to be revised. Appropriate explanatory text should also be added to clarify its content.
3. There is a practical concern regarding the applicability of WAG-based methods. These approaches require access to at least two models, whether benign or poisoned. This is a much stronger assumption than having access to a small clean dataset, and it significantly limits the feasibility of applying such methods in real-world scenarios.
4. I acknowledge that the search process only needs to be performed once to be effective. However, according to the experimental results, this process incurs a considerable computational cost, taking approximately six hours. This may even exceed the time required to train the model. For users with limited computational resources, such an overhead is significant and cannot be overlooked. The authors are advised to consider optimizing the search procedure to enhance its practicality.
5. According to the results presented in Table 3, when three models are fused, the performance of WAG and the proposed method is nearly identical. This raises an interesting question: would the performance gap become even smaller if more models were used? Furthermore, does this suggest that there is no fundamental advantage between weight aggregation and module switching?

---

> ### Author Rebuttal · Authors · 2025-07-31
>
> We thank the reviewer for their insightful feedback. Below, we address each point in detail.
>
> ---
>
> **W1:** The method assumes a two-stage backdoor training process, where models are first pre-trained on clean data and then poisoned. However, real-world attacks may train from scratch without a benign phase. Can the authors clarify how their method performs under such circumstances and provide experimental results under this setting?
>
> **A1:** We appreciate this insightful question. To our knowledge, injecting backdoors during the fine-tuning stage is the more common practice, as pretrained models are typically released by reputable organizations and are generally assumed to be clean. This assumption is also adopted in most existing backdoor attack studies and is commonly used in defense research as well.
>
> While we acknowledge that training from scratch with backdoor injection is theoretically possible, we are not aware of prior work adopting this setting for large pretrained models, likely due to the substantial computational cost. Moreover, we were unable to fully simulate this setting due to resource limitations. To approximate it, we consider a two-stage training process inspired by [22] : (1) we first fine-tuned a pretrained model on a backdoored IMDB dataset; (2) we then further fine-tuned this model on clean SST-2 data. We applied both WAG and our method to defend against such a model. The results are shown below:
>
> | No. | Setting                       | SST-2 CACC | SST-2 Badnet ASR | SST-2 InsertSent ASR |
> | --- | ----------------------------- | ---------- | ---------------- | -------------------- |
> | 1   | Pretrained → IMDB-Badnet      | 88.6       | 91.8             | -                    |
> | 2   | 1 → SST-2 Clean               | 96.4       | 83.8             | -                    |
> | 3   | Pretrained → SST-2 InsertSent | 96.3       | -                | 100.0                |
> | 4   | Merge 2 & 3 (WAG)             | 96.9       | **4.3**             | 6.3                  |
> | 5   | Merge 2 & 3 (Ours)            | 96.2       | 4.5          | **5.0**              |
>
> We observe that both WAG and our method achieve strong defense performance. We attribute this to the second clean fine-tuning stage weakening the backdoor effect, making it easier to defend. These results provide insight into our method's robustness in such a simulated setting.
>
> [22] Kurita et al. Weight poisoning attacks on pre-trained models. (ACL 2020)
>
> ---
>
> **W2:** The presentation of Figure 3 is somewhat confusing and needs to be revised. Appropriate explanatory text should also be added to clarify its content.
>
> **A2:** Thank you for this helpful feedback. Figure 3 is intended to provide an intuitive illustration of a possible switched combination between two models, based on the rules introduced just above the figure. The red and blue nodes represent weight modules in the merged model, indicating their source. The origin of each module is determined by both intra-layer and cross-layer connections. The arrows depict the feedforward paths followed during model inference.
>
> To improve the clarity of the figure, we plan to revise both its visual structure and accompanying explanation. Specifically, we plan to:
> * (1) sketch the structure of a Transformer model, including the self-attention and feedforward (FFN) blocks and their associated weight parameters;
> * (2) clarify the notation used for the six modules in the current figure—for example, Q and K refer to $W_{\text{query}}$ and $W_{\text{key}}$ from the self-attention block, while I and P correspond to the input and output weights of the FFN (MLP) block;
> * (3) annotate examples of each switching rule directly in the figure, such as adding explanatory text for colored Q and K nodes to illustrate the Type 1 intra-layer adjacency penalty.
>
> We would also be grateful for any additional suggestions the reviewer may have to further improve the figure's clarity and presentation.
>
> ---
>
> **W3:** The WAG-based method relies on access to at least two models (benign or poisoned), which is a stronger assumption than having a small clean dataset. It significantly limits the feasibility of applying such methods in real-world scenarios.
>
> **A3:** We would like to clarify the practicality and feasibility of our method from four aspects: (1) Standard practice in model merging research, (2) Domain-related merging is feasible, (3) Real-world model availability, and (4) No assumption of clean models.
>
> First, our method follows a widely adopted setting in prior model merging work, such as WAG [2] and DAM [60], where models with the same architecture are assumed to be available to enable training-free knowledge transfer [19,33] or for use in multiple expert-merging scenarios [A,B]. We will add additional references to the model merging literature in the related work section.
>
> Second, to enhance practicality, the WAG paper (Table 4) also demonstrates that models trained on related domains can be effectively merged. For example, IMDB, Yelp, Amazon reviews, and SST-2—all involving movie or product reviews—can be combined despite stylistic differences. We evaluate our method under a similar domain-relevant setting by merging two poisoned models (IMDB-Badnet and SST-2 InsertSent), and observe strong defense performance:
>
> | No. | Setting                       | SST-2 CACC | SST-2 Badnet ASR | SST-2 InsertSent ASR | AVG. ASR |
> | --- | ----------------------------- | ---------- | ---------------- | -------------------- | -------- |
> | 1   | Pretrained → IMDB-Badnet      | 88.6       | 91.8             | -                    | -        |
> | 2   | Pretrained → SST-2 InsertSent | 96.3       | -                | 100.0                | -        |
> | 3   | Merge 1 & 2 (WAG)             | 95.5       | **18.3**         | 57.2                 | 37.8     |
> | 4   | Merge 1 & 2 (Ours)            | 95.5       | 19.5             | **33.3**             | **26.4** |
>
> Third, structurally compatible models are often accessible in practice. For instance, Hugging Face hosts over 100 `roberta-large` models in the sentiment domain, including 82 tagged as `sentiment`, 28 as `sst2`, and 13 as `imdb`, making model compatibility feasible in real-world scenarios.
>
> Finally, our method does not assume that either model is benign. It remains effective even in a challenging yet more practical setting where both source models may be compromised.
>
> [A] He et al. Merging Experts into One: Improving Computational Efficiency of Mixture of Experts. (EMNLP 2023)
> [B] Tang et al. Merging Multi-Task Models via Weight-Ensembling Mixture of Experts. (ICML 2024)
>
> ---
>
> **W4:** Although the search is performed once, it takes around six hours—exceeding the time required to train a model. With limited computational resources, such overhead is significant and cannot be overlooked. Suggest optimizing the search procedure to enhance its practicality.
>
> **A4:** We would like to highlight that the evolutionary search is a one-time, offline cost per architecture, not per task or deployment. For example, the unified strategy found (Fig. 6) generalizes well across different model types (e.g., RoBERTa, BERT, DeBERTa, ViT), multiple tasks (SST-2, MNLI, AG News), and varying poisoning rates (20%, 10%, 1%) and attack methods. We believe that even for a new architecture, several hours of offline computation is a reasonable cost for practical deployment. We also plan to release the strategies discovered as public artifacts to support ongoing research on backdoor defense.
>
> Moreover, the search runs entirely on CPU and does not require high-end GPUs, making it accessible to users who cannot afford finetuning but still aim to perform secure inference. In scenarios where the data is poisoned, training a new model alone may not suffice, and identifying or removing the malicious data can be an even more computationally expensive process.
>
> We acknowledge that the search procedure could be further optimized and view this as a valuable direction for future work.
>
> ---
>
> **W5:** Table 3 shows that the performance of WAG and the proposed method is nearly identical when fusing three models. Would the gap shrink further with more models? Does this suggest that module switching offers no fundamental advantage over weight aggregation?
>
> **A5:** Thank you for this observation. It is a well-documented phenomenon in both WAG [2] and DAM [60] that as the number of accessible models increases, the defense performance of model merging methods improves.
>
> However, a common constraint in prior methods is the reliance on 3–6 models to achieve strong defense performance. When only two models are available, the attack success rate (ASR) often remains high. This trend is evident in Tables 2 and 5 of WAG [2], and Tables 2 and 3 of DAM [60], where performance improves significantly with 4–6 models but remains weaker with only two.
>
> In addition, some methods designed for the two-model case—such as [29]—require one of the models to be clean. Otherwise, new vulnerabilities may arise, as discussed in [C] (Table 4).
>
> Our work is motivated by this gap. We aim to provide an effective defense using only two models, without assuming either is benign. This makes our method more practical and deployable for end users, especially in constrained settings.
>
> [C] He et al. TUBA: Cross-Lingual Transferability of Backdoor Attacks in LLMs with Instruction Tuning. (ACL 2025)

---

> > ### Comment · Reviewer_Jnww · 2025-08-03
> >
> > Most of my concerns have been satisfactorily addressed.
> >
> > However, I still have some doubts regarding the last issue. As the authors mentioned, in multi-model merging scenarios, the defense performance of WAG may not be inferior to that of the proposed method, and WAG has the additional advantage of lower overhead (requiring only aggregation). Therefore, I suggest the authors emphasize the intended use case of their method in the revised version, possibly by focusing on the two-model merging scenario.

---

> > > ### Author Response · Authors · 2025-08-03
> > >
> > > Thank you again for your valuable feedback and the time spent reviewing our paper. We’re glad to hear that most of your concerns have been satisfactorily addressed.
> > >
> > > Regarding the intended use case of our method, we plan to emphasize three key points in the revision to better motivate our work:
> > >
> > > 1. **Working with fewer models is a more practical setting.**
> > >
> > >    Reducing the number of required models better supports low-resource users and is common in real-world scenarios, as having two models requires less data and computation than many, and is often more feasible in domains with limited open-source models or data.
> > >
> > > 2. **Existing model combination methods could be less effective in two-model settings.**
> > >    Existing methods in two-model scenarios face two key constraints that may limit their effectiveness:
> > >
> > >    * the backdoor effect often remains noticeable, leaving room for improvement; and
> > >
> > >    * new vulnerabilities may arise if the reference model is compromised.
> > >
> > >    Our method is designed to mitigate both types of threats and provides an interpretable perspective on disrupting backdoor shortcuts, which may benefit future defense research.
> > >
> > > 3. **Merging more models potentially introduces greater risks**, including:
> > >
> > >    * a higher chance of unintentional transfer of new backdoor behavior; and
> > >
> > >    * a higher likelihood of *collusive attacks*, where multiple models—poisoned using the same method—encode similar malicious behavior, which may further undermine the effectiveness of weight aggregation.
> > >
> > >    Keeping the number of merged models to a minimum (e.g., two) helps mitigate these risks and reduces the associated auditing effort.
> > >
> > > We will incorporate these points into the revision to highlight the motivation and intended use cases of our method. Please feel free to let us know if we can assist with any further clarification.

---

### Official Review · Reviewer_DvkL · 2025-07-02

**Clarity:** 2
**Significance:** 2
**Originality:** 2
**Rating:** 4
**Confidence:** 4

**Summary:**

In defense of the rising concerns of backdoor attacks on DNNs, the authors of this paper proposed Module Switching. Their method is designed to selectively exchange network modules among models trained on other related domains. The authors then conducted experiments in both the text and vision application domains, demonstrating effectiveness.

**Questions:**

- Across different application domains, only transformer-based models are used (RoBERTa, BERT, ViT). There have been backdoor attacks proposed on other neural networks, such as CNNs and MLPs. Recent backdoor attacks have been proposed on different architectures like these, but the effect of the proposed defense is limited to transformers. Can the authors please provide some insights of the proposed method on other model architectures?

**Ethical Concerns:**

["NO or VERY MINOR ethics concerns only"]

**Final Justification:**

The authors addressed my concerns. Thus, I raised my rating. However, Reviewer utYj and Reviewer uUUc still have several concerns (e.g., unrealistic assumption) that are critical.  I hope the authors can also resolve these issues in the revision.

**Limitations:**

Yes

**Quality:**

2

**Strengths And Weaknesses:**

Strengths:
- The paper is well-motivated and theoretically sound. The mathematical details of the method shown in sections 4.2 and 4.3 demonstrate the rules of module combinations searching vigorously.
- The proposed method works across both text and image tasks, only with a single switch, found through an “evolutionary search algorithm” without re-searching.

Weaknesses:
- The pseudocode algorithms are formatted poorly and are not easy to read/follow. Please consider cleaning up the pseudocode.
- The experiment settings seem a bit outdated. Can the authors please provide justifications for the experimental selections, including model selection and other backdoor attacks examined/compared?
- Experiments can be enriched. For example, is there a reason why the authors only choose one model for the vision experiments while having three for the text experiments?

---

> ### Author Rebuttal · Authors · 2025-07-31
>
> We thank the reviewer for their insightful feedback. Below, we address each point in detail.
>
> ---
>
> **W1:** The pseudocode algorithms are formatted poorly and are not easy to read/follow. Please consider cleaning up the pseudocode.
>
> **A1:** Thank you for this feedback. We provide clarification on the purpose and logic of the two algorithms below.
>
> **Algorithm 1:** This algorithm assigns a model index (e.g., 1 or 2) to each module position in a transformer model. For two RoBERTa-large models (24 layers × 6 modules), this results in 144 positions.
>
> * **Lines 4–8:** Randomly initialize a population of candidates (e.g., 100), each with a full index assignment. Evaluate each using heuristic fitness.
>
> * **Lines 9–19:** Run evolutionary search for a fixed number of generations (e.g., 2 million): (1) select parents via tournament selection [41], (2) generate children through random mutation, and (3) retain the top-performing candidates (lines 16–17).
>
> **Algorithm 2:** This algorithm identifies the cleaner model by comparing feature distances to a WAG-optimized dummy input for a suspect class.
>
> * **Lines 5–14:** For each candidate class: line 7 optimizes a dummy input so the model predicts it as the target class; line 8 extracts its hidden representation; the mean cosine distance to clean samples from other classes is then computed. The class with the highest average distance is selected (line 14).
>
> * **Lines 15–20:** Compute each switched model's feature distance to the WAG dummy input for the suspect class to select the cleaner model.
>
> We would also appreciate it if the reviewer could highlight any specific lines that were unclear, as this would help us improve the presentation further.
>
> ---
>
> **W2:** The experiment settings seem a bit outdated. Can the authors please provide justifications for the experimental selections, including model selection and other backdoor attacks examined/compared?
>
> **A2:** We chose the models and backdoor attacks primarily based on the experimental setup of the baseline method WAG [2], to ensure a fair and controlled comparison. While WAG focuses on the text domain, we extend the evaluation to the vision domain to assess the broader robustness and generalizability of our method.
>
> To validate the relevance of our selections, we conducted an up-to-date survey of recent top-tier venues. We found that both the models and backdoor attacks used in our study remain commonly adopted in the latest literature, supporting the rationale behind our design.
>
> In the text domain, 4 out of 5 recent papers [2], [3], [68], [A], [B] adopt the same model family as ours, and all 5 study the same backdoor attacks—BadNet, InsertSent, and HiddenKiller—which we also include in our experiments.
>
> For the vision domain, we selected the ViT model as it belongs to the same architectural family as our text encoder, allowing us to test the transferability of our strategy across modalities. Recent studies [60] and [F] also evaluate backdoor attacks on Vision Transformer models. The attacks we study, such as BadNet and WaNet, are widely used in recent literature [C]–[F], reaffirming their relevance.
>
> [2] Arora et al. Here's a Free Lunch: Sanitizing Backdoored Models with Model Merge. (ACL 2024)
> [3] Yi et al. BadActs: A Universal Backdoor Defense in the Activation Space. (ACL 2024)
> [60] Yang et al. Mitigating the Backdoor Effect for Multi-Task Model Merging via Safety-Aware Subspace. (ICLR 2025)
> [68] Zhao et al. Defense Against Backdoor Attack on Pre-trained Language Models via Head Pruning and Attention Normalization. (ICML 2024)
> [A] Chen et al. PKAD: Pretrained Knowledge Is All You Need to Detect and Mitigate Textual Backdoor Attacks. (EMNLP 2024)
> [B] Zhang et al. BadWindtunnel: Defending Backdoor in High-Noise Simulated Training with Confidence Variance. (ACL 2025)
> [C] Xu et al. BAN: Detecting Backdoors Activated by Adversarial Neuron Noise. (NeurIPS 2024)
> [D] Zhu et al. Breaking the False Sense of Security in Backdoor Defense Through Re-Activation Attack. (NeurIPS 2024)
> [E] Hu et al. BBCaL: Black-box Backdoor Detection Under the Causality Lens. (ICLR 2025)
> [F] Hu et al. A Closer Look at Backdoor Attacks on CLIP. (ICML 2025)
>
> ---
>
> **W3:** Experiments can be enriched. For example, is there a reason why the authors only choose one model for the vision experiments while having three for the text experiments?
>
> **A3:** We would like to clarify that our submission already includes 9 experiments—5 main experiments and 4 ablation studies—carefully designed to be comprehensive and support different aspects of our claims.
>
> In the text domain, we use RoBERTa-large as the primary model and include two additional models in the ablation studies to demonstrate that our method generalizes across architectures. The vision experiment focuses on generalize the finding to another modality.
>
> To further enhance the experiment on vision models, during the rebuttal period, we evaluated ViT-large (24 layers) using the same strategy as in Figure 6. The results are shown below:
>
> | Dataset  | Combination     | Method     | CACC | Atk1 ASR | Atk2 ASR | AVG. ASR |
> | -------- | --------------- | ---------- | ---- | -------- | -------- | -------- |
> | CIFAR-10 | BadNet + BATT   | no-defense | 96.6 | 96.0     | 99.9     | 98.0     |
> |          |                 | WAG        | 98.1 | 0.5      | 0.1      | 0.3      |
> |          |                 | Ours       | 98.0 | **0.3**  | 0.1      | **0.2**  |
>
> The results are consistent to the claim for the advantage of our methods.
>
> We hope this clarifies our experimental choices and demonstrates the broad applicability and robustness of our method.
>
> ---
>
> **Q4:** The method is only evaluated on transformer-based models (BERT, RoBERTa, ViT). Can the authors please provide some insights on the proposed method for other model architectures, such as CNNs or MLPs?
>
> **A4:** We focus on transformer models for two reasons: (1) our primary baseline method demonstrates backdoor defense capability in the text domain using a transformer model, and we retain the same setting to validate our claims; (2) we further demonstrate that our method has cross-modality effectiveness, and transformers are the architecture showing strong performance across modalities ([G], [H]), making them suitable for this evaluation.
>
> While we acknowledge the lack of evaluation on CNNs and emerging architectures, we have explicitly mentioned this point in the Limitations section of our submission and regard it as a valuable future research direction. We are willing to share some insights on this matter.
>
> As discussed in Section 4.2 of our submission, we consider backdoors to be mainly transmitted through both feed-forward and residual connection paths, which are essential components of most modern neural networks (NNs). Consider a general NN expression:
>
> $
> y = F_{W_n}(\cdots F_{W_2}(F_{W_1}(x_0) + x_0) + x_1 \cdots ) + x_{n-1}, \quad x_i = F_{W_i}(x_{i-1}) + x_{i-1}
> $
>
> We consider the key insight to be preventing adjacent weight modules from originating in the same model. In the context of CNNs and MLPs, we believe it is worth exploring either layer-level disruption or even finer-grained kernel-level decomposition, which we leave for future investigation.
>
> [G] Dosovitskiy et al. An image is worth 16x16 words: Transformers for image recognition at scale. (ICLR 2021)
> [H] Khan et al. Transformers in vision: A survey. (ACM Computing Surveys 2022)

---

> ### Comment · Reviewer_DvkL · 2025-08-05
>
> I want to thank the authors for the response. The authors partially addressed my concerns (e.g., W1 and Q4). However, for W2 and W3, I still believe that the authors should consider, especially for the evaluation on the vision tasks, including more recent attacks (e.g., BATT[1], AdaptivePatch[2]) in the later version. Such a practice will strengthen the overall quality of the paper.
>
> [1]. BATT: BACKDOORATTACKWITHTRANSFORMATION-BASEDTRIGGERS -ICASSP 2023
>
> [2]. REVISITING THE ASSUMPTION OF LATENT SEPARABILITY FOR BACKDOOR DEFENSES - ICLR 2023

---

> > ### Author Response · Authors · 2025-08-07
> >
> > We thank the reviewer for the follow-up comments and helpful suggestions.
> >
> > **Regarding BATT [1]**: BATT is already included in our original submission (Tables 2, 10, and 11), where our method consistently demonstrates stronger defense performance across all settings.
> >
> > **Regarding AdaptivePatch [2]**: We appreciate the suggestion and have conducted additional experiments. Paper [2] proposes two adaptive variants—Adaptive-Blend and Adaptive-Patch. As shown in their Table 1, Adaptive-Patch consistently achieves higher ASR across all settings, so we selected it as the representative attack (which also aligns with the reviewer's recommendation) and adapted it to the ViT-Base setting during this short rebuttal period. Following the same transferability strategy used in our Figure 12, the results are shown below:
> >
> > | Setting | Attacks | CACC | ASR (Atk1) | ASR (Atk2) | ASR (Avg) |
> > | ---------- | ---------------- | ---- | ---------- | ---------- | --------- |
> > | No Defense | BadNet | 98.4 | 97.3 | – | – |
> > | No Defense | BATT | 98.4 | 99.4 | – | – |
> > | No Defense | Adaptive Patch | 98.0 | – | 86.1 | – |
> > | WAG | BadNet + Adap-Patch | 98.5 | 24.3 | 17.6 | 20.9 |
> > | Ours | BadNet + Adap-Patch | 98.5 | **13.9** | **16.5** | **15.2** |
> > | WAG | BATT + Adap-Patch | 98.7 | **0.8** | 10.3 | **5.5** |
> > | Ours | BATT + Adap-Patch | 98.6 | 1.3 | 10.3 | 5.8 |
> >
> > As shown, our method consistently demonstrates strong defense performance against both Adaptive-Patch and BATT attacks. We hope these results sufficiently address the reviewer’s concerns.

---

> > > ### Comment · Reviewer_DvkL · 2025-08-07
> > >
> > > I want to thank the authors for the prompt response to the follow-up question. I believe that more experimental results can indeed improve the quality of the paper, and I decided to raise my rating. However, please also address other reviewers' concerns, which are quite critical.

---

> > > > ### Author Response · Authors · 2025-08-09
> > > >
> > > > Dear Reviewer DvkL,
> > > >
> > > > We are truly grateful for your reassessment and decision to raise the overall rating of our paper. Your encouragement and constructive feedback mean a great deal to us, and we appreciate the time and effort you have invested in reviewing our work. We will incorporate your suggestions and also address the concerns raised by the other reviewers to further strengthen the quality and clarity of the paper.
> > > >
> > > > Best regards,
> > > >
> > > > The Authors

---

### Official Review · Reviewer_utYj · 2025-07-03

**Clarity:** 2
**Significance:** 2
**Originality:** 3
**Rating:** 4
**Confidence:** 4

**Summary:**

This paper proposes a backdoor defense strategy that disrupts the backdoor shortcut by switching and recombining modules across multiple backdoored models to construct a clean model. The authors define multiple module switching rules and employ an evolutionary algorithm to search for a candidate pool of switching strategies. The optimal strategy is then selected based on feature-space distance to effectively remove backdoors. The authors provide theoretical analysis for their method and conduct experiments to show the effectiveness of their design.

**Questions:**

1. Can the proposed method work effectively when backdoored models intended for switching have different target classes? Could the authors provide additional experimental results for this more realistic scenario?
2. Does the proposed method remain effective when backdoored models for switching are trained using the same type of backdoor attack?
3. Could the authors provide a detailed analysis of the computational cost for the proposed method?

**Ethical Concerns:**

["NO or VERY MINOR ethics concerns only"]

**Final Justification:**

The authors' rebuttal has partially addressed my main concerns. Accordingly, I have raised my original rating score.

**Limitations:**

yes

**Paper Formatting Concerns:**

/

**Quality:**

3

**Strengths And Weaknesses:**

Strengths
1. The proposed method provides an interesting perspective on backdoor defense.
2. The method achieves better experimental results compared to previous approaches that mitigate backdoors through direct weight averaging.
3. The paper provides theoretical analysis for the proposed method.

Weaknesses
1. Some key designs are questionable. The Suspect-class Detection mechanism proposed in Section 4.4 appears to be effective only when backdoored models share the same target class. However, in real-world scenarios, different backdoored models are unlikely to have identical target classes, as these are typically chosen by different attackers for their specific purposes. The authors should clarify this design and its practical implications.
2. The evaluation scope is limited. All experiments focus on switching modules between models trained with different types of backdoor attacks, without evaluating effectiveness when models are trained with the same attack type.
3. The computational cost analysis is insufficient. The method described in Section 4.4 appears computationally intensive, yet the paper lacks adequate analysis of the computational overhead. The paper should include a thorough cost analysis for the entire defense mechanism.

---

> ### Author Rebuttal · Authors · 2025-07-31
>
> We thank the reviewer for their insightful feedback. Below, we address each point in detail.
>
> ---
>
> **W1 & Q1:** The suspect-class detection mechanism in Section 4.4 assumes that backdoored models share the same target class—an assumption that may not hold in real-world scenarios. Can the proposed method still work effectively when the backdoored models intended for switching have different target classes? Could the authors provide additional experimental results for this more realistic setting?
>
> **A1:** Yes, our method remains effective in target-label-inconsistent scenarios, where the involved models have different target classes. We conducted additional experiments for both text and vision domains, where our method remains competitive with or outperforms the WAG baseline, as shown below:
>
> | Dataset  | Combination                             | Method | CACC | Atk1 ASR | Atk2 ASR | AVG. ASR |
> | -------- | --------------------------------------- | ------ | ---- | -------- | -------- | -------- |
> | SST-2    | BadNet (label=0) + InsertSent (label=1) | WAG    | 96.4 | 47.2     | 71.2     | 59.2     |
> |          |                                         | Ours   | 96.2 | **20.0** | **65.9** | **43.0** |
> | CIFAR-10 | BadNet (label=1) + BATT (label=2)       | WAG    | 98.7 | 0.3      | 18.8     | 9.6      |
> |          |                                         | Ours   | 98.6 | 0.4      | 19.1     | 9.8      |
>
> These results confirm that our method is robust even when the backdoored models differ in their target labels.
>
> ---
>
> **W2 & Q2:** The experiments do not evaluate cases where models are trained using the same backdoor attack. Does the proposed method remain effective when the backdoored models used for switching are generated with the same attack type?
>
> **A2:** Yes, our method works much better than WAG, even when switching between models attacked by same method with non-identical labels. The result are shown below:
>
> | Dataset | Combination                         | Method | CACC | Atk1 ASR | Atk2 ASR | AVG. ASR |
> | ------- | ----------------------------------- | ------ | ---- | -------- | -------- | -------- |
> | SST-2   | BadNet (label=0) + BadNet (label=1) | WAG    | 96.3 | 78.0     | **2.4**  | 40.2     |
> |         |                                     | Ours   | 96.1 | **38.7** | 16.0     | **27.4** |
>
> For cases that both switched model attack by same method with identical label, the significant bias cannot be fixed. However, the issues was also a limitation for WAG, as averaging or switching yields no change when models encode same bias. Analogously, exchanging ideas between individuals who share the same misconceptions may not lead to correction, but rather reinforce the existing viewpoint.
>
> We believe that the increasing diversity in datasets may help reduce such cases, as it lowers the likelihood of models sharing identical biases.
>
> ---
>
> **W3 & Q3:** The method described in Section 4.4 appears computationally intensive, yet the paper lacks adequate analysis of the computational overhead. Could the authors provide a thorough cost analysis for the entire defense mechanism?
>
> **A3:** We provide a breakdown of the computational cost for the two main components of our method:
>
> **Switched Model Selection (Section 4.4):** This step is computationally efficient due to the small number of data points involved. Specifically, Lines 7 and 8 of Algorithm 2 operate on a single dummy input, and Line 9 processes only 20–50 clean data points. We measured the total runtime on SST-2 and CIFAR-10 using a single NVIDIA RTX 4090 GPU:
>
> * **SST-2:** 3.83 seconds total, with Lines 5–13 taking 2.3 seconds
> * **CIFAR-10:** 43.08 seconds total, with Lines 5–13 taking 20.63 seconds
>
> Although runtime may vary slightly with the number of classes, it remains under minute level which is a minor computational burden.
>
> **Evolutionary Strategy Search (Section 4.3):** As reported in Section 5.1, the one-time search process takes approximately 6 hours for a 24-layer model. However, the search only needs to be performed once per architecture, as the resulting strategy is transferable across models (Table 16) and datasets (Table 7-9). Thus, end users can directly use the released strategy without incurring this cost.
>
> Importantly, the search runs entirely on CPUs without requiring high-performance hardware such as GPUs, which makes the design accessible and computationally feasible. We welcome any suggestions from the reviewer, which are greatly appreciated and helpful in improving our work.

---

> > ### Comment · Reviewer_utYj · 2025-08-05
> >
> > The authors have addressed my main concerns through additional experiments and clarifications. However, I recommend performing further experiments to fully resolve the issues raised in Q1 and Q2, and incorporating those results into the main manuscript.

---

> > > ### Author Response · Authors · 2025-08-05
> > >
> > > We thank the reviewer for the insightful comments. During the short rebuttal period, we conducted additional experiments addressing Q1 and Q2, including: (1) models with different target labels, (2) models attacked using the same backdoor method, and (3) cases from both text and vision domains. Across all settings, our method remains competitive with or outperforms the WAG baseline, consistent with our main findings. We will incorporate these results into the revision.
> > >
> > > We would also appreciate further guidance on which specific ablation settings would be considered sufficient to fully address the concerns. If the current settings are already deemed satisfactory, we are grateful for the positive assessment and a higher score.

---

### Author Response · Authors · 2025-08-09

Dear Reviewers and ACs,

We sincerely thank all reviewers for their time, valuable feedback, and constructive suggestions, which have helped us strengthen the paper.

As the author–reviewer discussion period nears its end, we would take this opportunity to summarize the key points raised by the reviewers and the main takeaways from our rebuttal.

* **Strengths recognized by reviewers**

  * The method tackles an important backdoor defense problem (*uUUc*) from an interesting perspective (*utYj*, *uUUc*) by breaking learned backdoor shortcuts via merging backdoored models (*uUUc*).
  * The approach is well-motivated and theoretically sound (*DvkL*). Rigorous analysis is presented (*utYj*, *DvkL*), demonstrating the superiority of the proposed method over the baseline and supporting this with extensive experiments (*Jnww*).
  * The method works across text and image tasks, and the strategy obtained via search transfers without re-searching (*DvkL*).

* **The addressed questions during the rebuttal**

  * We have addressed the experimental scope concern through a literature relevance review and additional experiments, and consequently Reviewer *DvkL* has verbally noted an increased rating. We added experiments to test generalizability: different model training strategies (*Jnww*), adaptive attack (*uUUc*), and label-inconsistent cases (*utYj*), further supporting the robustness of our approach across different settings.
  * We have addressed computational overhead with a detailed time breakdown and clarifications (*utYj*, *Jnww*).
  * We have clarified practicality and feasibility of model merging methods (e.g., WAG) with literature support, statistics, and added experiments (*Jnww*).
  * We have improved clarity: revision plans for Figure 3 and fewer-model motivation (*Jnww*), detailed pseudocode explanation (*DvkL*), and clarified synthetic network training and NLP dataset settings (*uUUc*).

* **An arguable point on the method’s generalizability, with our clarifications**

  * Broader range of model combinations (*utYj*): We added different-label and same-attack experiments; results remain consistent with our findings.
  * Cases where models may not preserve benign utility similarly (*uUUc*): We clarified that attackers typically maintain clean-data utility to avoid easy detection, and we tested the suggested layer-shuffling operation, finding that utility drops significantly and may not meet this objective. Defining the minimum required similarity between models remains an important future direction.
  * Other architectures beyond transformers (*uUUc*): We shared insights on applying the method to general neural networks (with satisfactory acknowledgement from *DvkL* on Q4). We have explored transformers and various synthetic MLPs with theoretical analysis, though we acknowledged CNNs and other emerging architectures could be discussed in future work.

We emphasize that demands for broader generalizability can become an open-ended exercise. Our work addresses representative and challenging scenarios, supported by theoretical analysis, and we have made substantial additional efforts during the rebuttal period. We hope these contributions and the overall comprehensiveness of our work will be taken into account. We are grateful to the reviewers and area chairs for their thoughtful evaluation and valuable time.

---

### Decision · Program_Chairs · 2025-09-17

**Decision:**

Reject

**Comment:**

This paper introduces a module-switching defense against backdoor attacks, offering an alternative to weight averaging by recombining modules from multiple compromised models with an evolutionary search strategy. Reviewers found the idea novel, well-motivated, and supported by theoretical analysis and strong empirical results across text and vision tasks. However, several concerns limited enthusiasm: the method assumes availability of models fine-tuned from the same backbone and often with shared target labels, raising questions about generality; evaluation was restricted to transformer architectures and a narrow range of datasets and attacks; computational overhead for strategy search is significant; and benign utility preservation is not theoretically guaranteed. While the rebuttal added experiments and clarifications that partially mitigated concerns, the overall generality and practicality remain insufficient for acceptance.